# A Critical Review on the Dosing and Safety of Antifungals Used in Exotic Avian and Reptile Species

**DOI:** 10.3390/jof9080810

**Published:** 2023-07-31

**Authors:** Naresh Visvanathan, Jolise Yi An Lim, Hui Ting Chng, Shangzhe Xie

**Affiliations:** 1Department of Pharmacy, National University of Singapore, Singapore 117559, Singapore; 2Mandai Wildlife Group, 80 Mandai Lake Road, Singapore 729826, Singapore

**Keywords:** antifungal, avian, efficacy, exotic species, fungal diseases, pharmacodynamic, pharmacokinetic, reptile, safety

## Abstract

Antifungals are used in exotic avian and reptile species for the treatment of fungal diseases. Dose extrapolations across species are common due to lack of species-specific pharmacological data. This may not be ideal because interspecies physiological differences may result in subtherapeutic dosing or toxicity. This critical review aims to collate existing pharmacological data to identify antifungals with the most evidence to support their safe and effective use. In the process, significant trends and gaps are also identified and discussed. An extensive search was conducted on PubMed and JSTOR, and relevant data were critically appraised. Itraconazole or voriconazole showed promising results in Japanese quails, racing pigeons and inland bearded dragons for the treatment of aspergillosis and CANV-related infections. Voriconazole neurotoxicity manifested as seizures in multiple penguins, but as lethargy or torticollis in cottonmouths. Itraconazole toxicity was predominantly hepatotoxicity, observed as liver abnormalities in inland bearded dragons and a Parson’s chameleon. Differences in formulations of itraconazole affected various absorption parameters. Non-linearities in voriconazole due to saturable metabolism and autoinduction showed opposing effects on clearance, especially in multiple-dosing regimens. These differences in pharmacokinetic parameters across species resulted in varying elimination half-lives. Terbinafine has been used in dermatomycoses, especially in reptiles, due to its keratinophilic nature, and no significant adverse events were observed. The use of fluconazole has declined due to resistance or its narrow spectrum of activity.

## 1. Introduction

Exotic animals are typically non-domesticated and non-indigenous to the geographical region they inhabit [1]. Fungal diseases are widespread and have led to the decline of multifarious exotic animals. Fatal dermatomycoses caused by *Nannizziopsis* spp. have been reported in captive reptiles, despite treatment with antifungals [2]. Dermatomycoses such as those caused by the fungal complex *Chrysosporium anamorph of Nannizziopsis vriesii* (CANV) are emerging infectious diseases in reptiles, with increasing prevalence and spread across geographical regions and, unfortunately, increasing antifungal resistance patterns [3]. Likewise, the most common fungal disease in avians, aspergillosis, has a high mortality rate [4]. Other than the avian anatomical and physiological characteristics which predisposes them to aspergillosis, azole resistance, especially in *Aspergillus fumigatus*, is a rising global concern requiring stricter epidemiological surveillance to reduce treatment failures [5]. Limited studies were collated for other fungal diseases, such as candidiasis and cryptococcosis, in avians. *Candida albicans* is the most notorious pathogen for candidiasis, while the pathogenic agents for cryptococcosis are *Cryptococcus neoformans* and *Cryptococcus gattii* [6]. Although some antifungals were studied for candidiasis and cryptococcosis, the sample sizes were not statistically significant for recommendations to be made. With rising mortality and morbidity rates in these exotic animals, there is a need to refine the currently inadequate armamentaria of antifungals used in the treatment of fungal diseases.

Antifungals are often used in the treatment of fungal diseases in exotic animals to eliminate fungal pathogens and resolve clinical symptoms. Different classes of antifungals such as azoles, allylamines, echinocandins, nucleoside analogues and polyenes with different mechanism(s) of actions have been used [7]. Veterinarians commonly adopt recommendations from formularies such as the Exotic Animal Formulary (EAF) to guide dosing regimens of antifungals. Due to the lack of pharmacological data in some exotic animals, such as those from the crocodylia or phoenicopteriformes order, veterinarians often extrapolate doses from closely-related or sometimes even distantly related species. This may not be ideal due to marked interspecies differences in metabolic rates and physiological features, even within closely-related species, which may affect the pharmacological profile of antifungals and ultimately therapeutic outcomes. When inappropriate dose extrapolations are made, suboptimal treatment outcomes may arise in the form of increased risk of toxicity, subtherapeutic dosing or treatment failure, subsequently contributing to the decline of exotic animals [8].

The pharmacology of antifungals is comprised of their pharmacokinetic (PK) and pharmacodynamic (PD) properties. PK is the study of how the body handles a drug, which is largely due to absorption, distribution, metabolism, and excretion processes [9]. Primary PK parameters such as clearance (CL) and volume of distribution (V) are controlled by physiological factors such as intrinsic metabolic enzymes, blood flow, protein levels and others [10]. These factors may vary across species, leading to variabilities in drug disposition which ultimately affect the exposure of the animal to the antifungal, which could either be sub-therapeutic or toxic. PD is the study of how drugs interact with the body to produce therapeutic and toxic effects [9]. Interspecies differences in receptor sub-types or the nature of tissues may thus result in differences in efficacy and safety [11]. The therapeutic outcomes of fungal diseases are fundamentally governed by this PK–PD interplay, which provides guidance on which antifungal is efficacious and/or safe in treating fungal diseases in exotic animals.

Therefore, this critical review aimed to collate the pharmacological data of antifungals across various exotic animals from existing literature to determine which antifungals have the most pharmacological data to support their effective and safe use and discuss significant trends or gaps in knowledge to provide directions for future studies. Our findings also provide caution in dose extrapolation and updated therapeutic recommendations for optimal antifungal treatment outcomes.

## 2. Materials and Methods

An extensive search was carried out on Embase, Google Scholar, JSTOR, PubMed and Web of Science databases using keywords such as antifungal, dosing, avian, reptile, pharmacokinetic, pharmacodynamic, efficacy and safety. A comprehensive list of search terms can be found in Appendix B (Table A1). The reference lists from extracted papers were also examined to identify any potentially relevant papers.

The following data were extracted from the collated papers if available: causative fungal pathogen(s) and their related fungal diseases, site(s) of infection, common and scientific name of species, number and health status of species, drug name and dosing, dosing interval and duration, route of administration, mean steady state plasma concentration (C_ss_) or peak plasma drug concentration (C_max_), time taken to reach C_max_ (T_max_), area under plasma concentration–time curve from dosing to the last measured time point (AUC_0–last_) or to infinity (AUC_0–∞_), Minimum Inhibitory Concentration (MIC), half-life (elimination or terminal), bioavailability, V, CL, reports of efficacy and adverse events. These data were critically appraised with reference to the EAF or other tertiary sources to achieve the aims of this critical review.

## 3. Key Interspecies Differences between Avians and Reptiles

### 3.1. Physiological Differences Impacting Susceptibility to Different Fungal Pathogens

Thermoregulation is one of the key differences between reptile and avian species. Reptiles are ectothermic as their body temperature is influenced by ambient temperatures. Poor husbandry with suboptimal temperature may impair their immune functioning, predisposing them to a diverse range of fungal pathogens [12,13]. In contrast, avians are endothermic and maintain a constant internal body temperature at around 35–40 °C [14]. This could explain their higher predisposition to aspergillosis, as *Aspergillus* spp. such as *Aspergillus fumigatus* thrive in warmer temperatures [15]. Due to the differing susceptibilities towards fungal pathogens, different antifungals are required to treat fungal diseases in exotic animals.

Reptiles have heavily keratinized epidermis compared to avians, making them more susceptible to formerly known keratinophilic *Chrysosporium anamorph of Nannizziopsis vriesii* (CANV) complex pathogens [3,16].

### 3.2. Physiological Differences in Relation to PK

Primary PK parameters that affect the disposition of antifungals are dependent on numerous physiological variables, which can subsequently affect secondary PK parameters such as C_max_, T_max_ or elimination half-life, which collectively influence antifungal dosing regimens [17].

The lower body temperatures in reptiles result in slower gastrointestinal motility [18]. Within reptiles, gut transit time increases with body size and is longer for herbivores than for carnivores [19]. Gastric pH is also known to vary from domestic birds, such as chickens, to facultative scavengers, such as red-tailed hawks. Consequently, these variabilities can have profound impacts on the absorption profiles of orally administered antifungals [20].

Fat compositions vary within avians due to dietary differences or migratory patterns [21] and within reptiles due to physiological states of feeding, fasting or vitellogenesis [22]. This can affect the distribution of antifungals, especially if they have high V, such as terbinafine, which can accumulate in the fatty tissues and slowly redistribute back to the blood, resulting in longer elimination half-lives and less frequent dosing intervals. Plasma protein levels may also vary across avians and reptiles, which also consequently influence V, and thus elimination half-life [23,24].

Higher metabolic rates in avians [25] increase drug metabolism, reflected by an increase in CL, which can decrease elimination half-life. Furthermore, elevations in ambient temperatures may also increase metabolic rates and affect drug metabolism [26]. The expression levels of CYP450 enzymes and genetic polymorphisms may differ across species, even across closely-related species [27], which could potentially cause variability in the disposition of azoles that are predominantly metabolised by CYP450 enzymes [28].

Differences in elimination half-lives explicate the need for different dosing regimens in avians and reptiles for optimal therapeutic outcomes. Furthermore, these various physiological differences have an unpredictable effect on PK and, thus, species-specific data would provide the most accurate recommendations for efficacious and safe antifungal dosing regimens.

## 4. Pharmacology of Antifungals

The major classes of antifungals in veterinary medicine are azoles, allylamines and polyenes, due to their ability to inhibit the formation of or directly interact with ergosterol, which is a critical component of fungal membranes [29]. The focus of this review will be on azoles and allylamines, which have had more extensive studies. A comprehensive list of the antifungals used can be found in the Appendix A.

Antifungal susceptibility testing (AST) is used to guide the selection of antifungals by determining MIC values for optimal therapeutic outcomes. MIC is a measure of the minimum drug concentration required to inhibit the growth of the pathogen, which indicates the susceptibilities of fungal pathogens to antifungals [30]. To predict treatment efficacy and safety, MIC is often correlated with PK parameters such as C_max_ or AUC over MIC or T > MIC, depending on the mode of action of the antifungal and whether or not they exhibit a post antifungal effect (PAFE) [31], and these are summarised in Table 1.

## 5. Pharmacodynamic Variabilities in Avians and Reptiles

Seventy studies or case reports on the efficacy and/or safety of antifungals in avians were found, with itraconazole (n = 20) and voriconazole (n = 17) being the most commonly used antifungal, especially in Psittaciformes and Sphenisciformes. In reptiles, 30 such reports were found, with itraconazole (n = 13), terbinafine (n = 5) and voriconazole (n = 5) being the most commonly used antifungal, especially in Squamata. An extensive summary of the efficacy and safety data for avians and reptiles is shown in Appendix A.

### 5.1. Evaluation of Dosing Regimens in Relation to Efficacy

Clinical resolution of lesions or symptoms are often used to assess the efficacy of antifungals in treating fungal diseases. Objective measures of efficacy include the use of MIC data to determine the susceptibility of fungal pathogens to antifungals, or to maintain therapeutic plasma concentrations above MIC. Determination of MIC of fungal isolates were limited and often relied upon comparisons with reported values in human studies or other related species. AST is recommended, when possible, to attain specific MIC values, especially for causative pathogens with MIC yet to be determined and also to look out for emerging resistance patterns.

#### 5.1.1. Type of Fungal Pathogens/Disease

The type of fungal pathogens is pivotal in the selection of antifungals to optimise treatment efficacy in exotic animals. The evaluation on the use of different antifungals for the more common fungal diseases in avian and reptiles are summarised in Table 2 and Table 3.

Aspergillosis commonly involves the respiratory tract, with *Aspergillus fumigatus* being the most common causative pathogen. Voriconazole appears to be efficacious in treating *Aspergillus fumigatus* infections in Japanese quails and racing pigeons (Table 2), and the large sample size provides further credibility. AST conducted on racing pigeons revealed an MIC of 0.25µg/mL for *Aspergillus fumigatus*, showing susceptibility to voriconazole, which could explain the efficacy observed. However, not all studies performed AST, and many used MIC data from other species to evaluate the efficacy of antifungals [32]. Additionally, *Aspergillus fumigatus* can be easily confused with more than 11 other *Aspergillus* spp. based on their conidial morphology [33], including *Aspergillus lentulus*, which is typically known to be resistant to triazoles and amphotericin B [34]. Accurate identification of the causative pathogen and its susceptibility to different antifungals is crucial to ensure efficacy in treatment. AST is recommended, especially for severely ill exotic animals, where time is critical and using antifungals that are resistant to the causative fungal isolates may result in poor clinical outcomes.

For *Aspergillus flavus* infections, itraconazole proved efficacious in turkeys by reducing the lung lesion score (Table 2). In Okinawa rails, however, though MIC revealed susceptibility of *Aspergillus flavus* to itraconazole, inefficacy was observed, possibly due to low concentrations in the air sacs. Voriconazole might also be efficacious for *Aspergillus flavus* infections, but this is inconclusive due to the small sample sizes of studies.

CANV infections in reptiles are the rising cause of mortality, especially in bearded dragons (n = 7) [35], which also have the most studies/case reports collated (Table 3). Numerous pathogens under the CANV denomination were identified and have been subsequently renamed to include three genera: *Nannizziopsis* spp., *Ophidiomyces* spp. and *Paranannizziopsis* spp. Formerly termed “yellow fungus disease” has been replaced by “nannizziomycosis” and “paranannizziomycosis”; thus, careful interpretation of older literature is warranted [3]. With the limited reports of antifungal therapy in CANV-related infections, itraconazole did not show efficacy in most reptiles, except the tuataras infected by *Paranannizziopsis* spp. (Table 3). In a direct comparison study, voriconazole demonstrated better efficacy in clearing CANV infection and had a higher survival rate (Table 3). In another giant girdled lizard, voriconazole used at the same dose cleared CANV infection and AST revealed the lowest MIC (0.25µg/mL) for voriconazole amongst the other antifungals tested [36]. Hence, voriconazole shows promising potential for the treatment of CANV infections, but further studies should be conducted in other reptile species to ensure the efficacy and safety of the dosing regimens [36].

Ophidiomycosis, previously termed “snake fungal disease” appears to only cause disease in snakes, though the reasons are unclear [3]. Studies on antifungal therapy for ophidiomycosis are scarce, with inconclusive efficacy (Table 3). Unconventional methods with terbinafine using subcutaneous implants and nebulization have been described to be possibly efficacious based on plasma-concentration studies, but further studies on actual diseased snakes are required to confirm these findings. Such methods are preferred, especially when venomous snakes are involved and also due to their bimonthly feeding, which makes oral administration of antifungals unfavourable [37]. Though voriconazole may also be a possible option, efficacy is unclear.

While MIC data is important for assessing efficacy, efficacy is not guaranteed, even when the pathogen is susceptible to the antifungal in vitro. Other reasons for the lack of efficacy may be insufficient concentrations at the site of infection, acquired resistance, interspecies differences affecting the disposition of antifungals, or poor condition of species at initial presentation. Such factors should also be considered when assessing the efficacy of antifungals.

#### 5.1.2. Site of Infection

The site of infection plays a crucial role in the selection of appropriate antifungals. Fungal diseases in reptiles commonly occur in the cutaneous layers, and their much thinner hypodermis can predispose them to subsequent invasion of deeper tissues, such as the bones or muscles, requiring systemic antifungals [38]. The common sites of infection in avians preferentially involve the respiratory tract, possibly because the primary route of fungal invasion is via inhalation [39].

Tissue concentrations of antifungals are more clinically relevant and reflective of the efficacy of the antifungal rather than the plasma concentrations, though they are not commonly measured in PK studies. A typical dose of itraconazole 10 mg/kg seems to be efficacious in different species, such as the African grey parrots and red-tailed hawks, but not in Okinawa rail or Pesquet’s parrots (Table 2). A possible reason for this could be that the tissue concentrations were not high enough to eliminate the fungal pathogens. In mallard ducks and racing pigeons, itraconazole concentrations in lung and brain tissues were the lowest, and it was posited that a significantly higher dose would be required to have a fungicidal effect in lung tissues [39,40]. Voriconazole tissue concentrations were not measured in any studies during our search. In reptiles, the keratinophilic nature of terbinafine makes it ideal for the treatment of dermatomycoses, due to accumulation in keratinized tissues [41].

Tissue concentration studies were scarce, and future studies should focus on methods to predict tissue concentrations with plasma concentrations in specific species to better assess the therapeutic efficacy of antifungals.

### 5.2. Evaluation of Dosing Regimens in Relation to Safety

Species-specific data are crucial when assessing the safety of antifungals. Itraconazole, voriconazole and terbinafine had more extensive safety data reported in avians and reptiles (Appendix A), which can be found in the Appendix A. Variable manifestations of neurotoxicity were observed for voriconazole in the form of lethargy, anorexia, ataxia and seizure-like symptoms in multiple penguin species such as Humboldt, African or Macaroni penguins, hind limb paresis in red-eared slider turtles and torticollis in cottonmouths, but not in African grey parrots or Japanese quails, despite using significantly higher doses (Appendix A). Possible hepatotoxicity from voriconazole administration was seen in the form of mild increases in aspartate aminotransferase in African grey parrots and a few inland bearded dragons (Appendix A). Similar adverse events have been observed in humans, and high trough concentrations are reported to increase the risk of neurotoxicity and hepatotoxicity [42], though the target trough concentrations in specific exotic species are unclear. Adverse events in itraconazole were observed as hepatotoxicity mainly in the form of increases in aspartate aminotransferase in bearded dragons and a Parson’s chameleon, but not in tuataras (Appendix A), despite being given similar doses. Terbinafine was generally found to be safe in both avians and reptiles. For the other antifungals, there were either no adverse events reported or insufficient sample sizes to evaluate safety accurately.

Though many single-dose studies reported no adverse events, safety cannot be accurately concluded as toxicity may sometimes only be observed upon chronic administration. This was evident in the numerous penguin species which showed neurotoxicity in multiple-dosing regimens but not in single-dosing regimens (Appendix A). Furthermore, the deranged physiology in immunocompromised animals may further predispose them to adverse events that healthy animals are not subjected to. Adverse drug reactions may be unpredictable and therefore hard to predict in specific species [43]. Hence, close therapeutic drug monitoring is recommended, especially for the azoles, to ensure safe dosing regimens.

**Table 2 jof-09-00810-t002:** Summary of efficacy and safety data in avian species accompanied by a critical analysis of the study findings.

Site of Infection	Antifungal Used	Dosing Regimen	Species	Evaluation of Dosing Regimens in Relation to Efficacy and Safety	References
***Aspergillus* spp.^a^**					
Respiratory tract	Clotrimazole Itraconazole	10 mg/mL NEB SID for 30–45 min on 3 day on–2 day off schedule PO 10 mg/kg SID × 1 month	African grey parrots (n = 1) Citron-crested cockatoo (n = 1) Gyrfalcon (n = 1) Red-tailed hawk (n = 1) Yellow-naped Amazon (n = 1)	Clotrimazole: • Efficacy observed for all species studied. Common mild adverse event of regurgitation for citron-crested cockatoo and Yellow-naped Amazon. Potential for clotrimazole nebulisation to be efficacious for aspergillosis in avians. Despite small sample size, efficacy was apparent for a range of avian species from different avian orders. Itraconazole: • Efficacy in terms of responsiveness to treatment. However, there was no specificity as to how efficacy was seen (e.g., resolution of lesions, improvement of symptoms, etc.). Hence, such efficacy data may be less reliable. Safety observed.	[44]
Respiratory tract	Amphotericin B Flucytosine	Intratracheal 1 mg/kg BID × 3 days IV 1.5 mg/kg TID × 3 days NEB 1 mg/mL of saline BID × 20 days PO 120 mg/kg QID × 20 days	Gyrfalcon (n = 1)	• Efficacy in terms of stabilisation of condition was not specific enough to accurately assess efficacy of amphotericin B in treating aspergillosis. Safety observed.	[44]
Respiratory tract	Amphotericin B	NEB 1 mg/mL of saline BID × 31 days	Yellow-naped Amazon (n = 1)	• Efficacy in terms of dramatic response to therapy was not specific enough to accurately assess efficacy of amphotericin B in treating aspergillosis. Safety observed.	[44]
NR	Clotrimazole	NEB 40 mg for 1 h a day	Micronesian kingfisher (n = 1)	• Efficacy and safety observed.	[45]
NR	Fluconazole	PO 15 mg/kg q12 h × 5–75 days	African grey parrots (n = 23)	• Efficacy and safety observed.	[46]
Eyelids, head	Itraconazole Miconazole	PO 15 mg/kg q12–24 h × 3 months Topical BID on lesions	Falcon peregrinus × Falcon rusticolus (n = 1)	• Efficacy and safety of itraconazole was observed when given PO 15 m/kg q12 h. Efficacy and safety of miconazole was not reported.	[47]
NR	Itraconazole	PO 10 mg/kg SD	Lesser flamingo (n = 17)	• Efficacy was inconclusive because it was a single-dose study. Multiple dosing should be conducted as they are more clinically relevant. Large sample size supports safety of itraconazole in lesser flamingo.	[48]
NR	Voriconazole	PO 12.5 mg/kg q12 h × up to 91 days	Gyrfalcons × peregrine falcon hybrid (n = 1) Saker falcons (n = 2)	• Efficacy was not reported. Safety was observed.	[49]
NR	Voriconazole	PO 20 or 40 mg/kg SD	Japanese quail (n = 38 for each dosing)	• Efficacy and safety were observed.	[50]
Respiratory tract	Amphotericin B	IV 0.5–0.75 mg	Prairie falcons (n = 2)	• Lack of efficacy might be due to critical condition for falcons at initial presentation, and not the inefficaciousness of amphotericin B.	[51]
Respiratory tract and hematogenous involvement	Flucytosine	PO 60 mg/kg BID	Crested wood partridges (n = 4)	• Lack of efficacy was observed. Safety was not reported.	[52]
^a^ Evaluation of antifungal therapy: • General site of infection was observed to involve the respiratory tract. Although only two papers were collated on clotrimazole nebulisation, the studies showed efficacy of the drug in a range of avian species. Together with only mild adverse events observed across species, clotrimazole might be a potential antifungal for treating respiratory aspergillosis. Use of clotrimazole has not been widely studied, hence more studies can be conducted to assess its efficacy and safety. Fluconazole showed efficacy and safety in African grey parrots and Falcon peregrinus and Falcon rusticolus. Large sample size in the study on African grey parrots further supports fluconazole’s efficacy and safety in treating aspergillosis in avian species. Site of infection was not reported. Voriconazole showed efficacy in the study on Japanese quails where a large sample size was used, but the site of infection was not reported. Voriconazole showed safe use in Japanese quails and falcon species. Hence, it is likely to be safe when used to treat aspergillosis in avian species. However, more studies are still needed to assess its efficacy and safety in other avian species. There were insufficient data to assess the efficacy and safety for itraconazole, amphotericin B, flucytosine and miconazole in avian species.					
** *Aspergillus flavus ^b^* **					
Lungs	Ketoconazole Enilconazole Itraconazole	PO 50 mg/kg PO 150 mg/kg Aerosol spray: 50 mg/kg PO 6 mg/kg	Turkeys (n = 20–40)	• Efficacy seen for PO itraconazole and enilconazole aerosol spray, where PO itraconazole was more efficacious because it resulted in the reduction of lesion score in both lungs and air sacs by more 90% and a lower re-isolation rate of fungi of 30% (trial 2). Enilconazole aerosol spray efficacy was seen in the reduction of lesion score in only the lungs (>50%), and re-isolation rate was 80–100%. PO ketoconazole and PO enilconazole were inefficacious. Safety was not reported.	[53]
Air sac	Voriconazole Micafungin Itraconazole	IM 23.5 mg/kg q12–24 h SC 11.6 or 22.7 mg/kg q12–24 h PO 10 mg/kg q12 h × 162 days	Okinawa rail (n = 1)	• Lack of efficacy was observed despite the use of these three antifungals. MIC testing was conducted, and the pathogen showed low susceptibility to voriconazole and micafungin. Safety was not reported. Likely reasons for the lack of efficacy of itraconazole, even though the fact that the pathogen showed susceptibility to itraconazole was not mentioned in the paper, might include the dose being too low or not high enough to reach the air sacs, the long-term nature of antifungal regimen might have resulted in antifungal resistance, or critical condition at initial presentation.	[54]
^b^ Evaluation of antifungal therapy: • Based on the study on 40 turkeys, itraconazole proved its efficacy in reaching the lungs and air sacs. Even though safety was not reported, there is evidence that itraconazole is potentially a recommended antifungal for treating *Aspergillus flavus* infections. Even though itraconazole was inefficacious in treating the air sac infection in Okinawa rail, only one subject was studied. Voriconazole may still be efficacious, but due to the small sample size and critical condition of the rail at initial presentation, it showed inefficacy in treating the infection. Enilconazole aerosol spray may be the next antifungal choice for treating *Aspergillus flavus* infection of the lungs in turkeys, but the route of administration is not as convenient as the oral route. However, oral enilconazole was seen to be inefficacious in the same study. Hence, the most appropriate antifungal for treating *Aspergillus flavus* infection of the lungs in turkeys would be PO itraconazole. Because safety was not reported, more studies could be done to evaluate its safe use so that better antifungal recommendations can be made. With only two papers on *Aspergillus flavus* infections, more studies should be conducted to assess the efficacy and safety of different antifungals, especially itraconazole, across different avian species.					
** *Aspergillus fumigatus ^c^* **					
Pectoral muscle	Itraconazole Miconazole Amphotericin B (liposomal formulation)	PO 20 mg/kg q12 h × 50 days Topical twice weekly Topical 1.35 mg/kg q24 h × 30 days → discontinued for 15 days → q24 h × 14 days → every 3/4 days × 16 days → every 5–7 days × 60 days	Goliath heron (n = 1)	• Lack of efficacy for both PO itraconazole and topical miconazole. Efficacy was observed when liposomally encapsulated amphotericin B was administered topically. Liposomal formulation of amphotericin B has reduced nephrotoxicity and longer duration of activity related to conventional formulation.	[55]
Trachea	Itraconazole	10% NEB SID for 30 min × 6 days 4% NEB SID for 30 min × 6 days	Japanese quail (n = 40)	• 10% suspension was more efficacious than 4% suspension. Safety was observed for both suspensions. 4% suspension was inefficacious, which is likely due to reduced deposition of itraconazole and hence reduced availability of parent drug for dissolution and metabolism to form the metabolite, hydroxy-itraconazole (OH-ITRA).	[56]
Respiratory tract	Itraconazole	PO 10 mg/kg/day × 7 days	Japanese quail (n = 60)	• Reduction in mortality may not be a good marker for evaluating the efficacy of antifungal in treating fungal disease. Safety was not reported.	[57]
Trachea	Voriconazole	PO 20 mg/kg SD PO 20 mg/kg q24 h PO 40 mg/kg SD PO 40 mg/kg q24 h	Japanese quail (n = 76 for single dose) (n = 40 for multiple dose)	• Single dose (40 mg/kg): plasma concentrations remained above 0.5 µg/mL (concentration where improved treatment success was reported) for longer duration (12 h). 40 mg/kg q24 h showed efficacy in terms of longer drug exposure, prolonged survival and fewer colony forming units, which was previously validated and discussed; safety was reported.	[50]
Ocular	Voriconazole	Topical 1 drop OS q4–6 h PO 20 mg/kg q12 h × 35 days	Khaki Campbell duck (n = 1)	• PO voriconazole showed better efficacy than topical formulation due to better penetration into the cornea through the presence of corneal vascularisation in this case. Mild toxicity was observed. PO voriconazole was seen to be able to reach the physiologically privileged site (eye) in this case, hence this dosing regimen is likely to be efficacious for ocular *Aspergillus Fumigatus* infection in this species.	[58]
NR	Voriconazole	PO 8 mg/kg SID → 5 mg/kg SID → 5 days on treatment followed by 2-day off period × several years	Magellanic penguin (n = 1)	• Safety was apparent when voriconazole was given via 5 days on–2 days off dosing regimen. Additionally, safety was observed even when this dosing regimen was used for several years, which shows the safety of voriconazole during long-term treatment. Multiple dosing or long-term dosing of antifungals are more clinically relevant. Efficacy was unclear. Because voriconazole has shown a range of variable toxicities across avian species, the use of n day on–n day off dosing intervals might be useful to avoid overdosing and toxicity issues.	[59]
Respiratory tract, air sac	Amphotericin B	IM 2.0 mg/kg SID	Peruvian penguins (n = 1)	• Lack of efficacy was observed. Safety was not reported.	[60]
Lung	Voriconazole	PO 10 mg/kg q12 h × 14 days PO 20 mg/kg q24 h × 14 days	Racing pigeons (n = 10 for each dosing)	• MIC = 0.25 µg/mL (isolate was susceptible to voriconazole). 10 mg/kg q12 h proved to be more efficacious than 20 mg/kg q24 h but toxicity was slightly more severe. Shorter dosing interval might provide sufficient or high enough plasma concentrations (>0.5 µg/mL) to achieve therapeutic efficacy.	[61]
^c^ Evaluation of antifungal therapy: • Liposomal amphotericin B might be efficacious for *Aspergillus Fumigatus* infection of the pectoral muscle in goliath heron but only one subject was studied. Itraconazole (NEB/PO) and PO voriconazole (40 mg/kg q24 h) were observed to be efficacious and generally safe in treating *Aspergillus Fumigatus* infection involving the respiratory tract in Japanese quails. PO voriconazole (10 mg/kg q12 h) was efficacious in treating infection involving the lungs in racing pigeons but mild toxicity was reported. The large sample size in these studies improves the credibility of such data. Higher dose was used in Japanese quail than in racing pigeons, but no toxicity was observed. This is likely due to the interspecies differences across different avian species, even across closely-related species. PO voriconazole might be a useful antifungal when *Aspergillus Fumigatus* infection involves physiologically privileged sites such as the eye. Efficacy was apparent in Khaki Campbell duck, but only one subject was studied. PO voriconazole was safe, even when long-term dosing was administered to magellanic penguin. Long-term dosing of antifungals is more clinically relevant in the treatment of fungal diseases. Use of n day on–n day off dosing regimen may be useful in reducing the risk of accumulation, overdosing and toxicity, especially because voriconazole has shown variable toxicity across different avian and reptile species.					
** *Candida albicans ^d^* **					
Respiratory tract	Nystatin Itraconazole Amphotericin B Fluconazole	PO 400,000 units/kg q12 h PO 10 mg/kg q12 h × 30 days Intralesional: 1 mg/kg NEB 1 mg/mL of saline for 15 min q8 h × 2 days PO 5 mg/kg q12 h × 1 month	Sun conure (n = 1)	• Bird improved clinically after initiating itraconazole and amphotericin B but repeat complete blood count 1 week later showed increased leucocytosis with increased heterophilia and monocytosis. Based on the hematologic results, and amid concerns of drug resistance, treatment regimen was changed from itraconazole to fluconazole. Isolated pathogen was susceptible to fluconazole, itraconazole and voriconazole. Fluconazole showed efficacy in terms of significant signs of improvement (increased appetite, normal body weight). Safety was not reported.	[62]
Gnathotheca, mandibular bone	Itraconazole	PO 10 mg/kg q24 h × 10 days	Lesser flamingo (n = 1)	• Lack of efficacy was observed. Safety was not reported.	[63]
Proventriculus	Ketoconazole	PO 10 mg/kg q12 h × 3 weeks	Lesser sulphur-crested cockatoo (n = 1)	• Efficacy was observed. Safety was not reported.	[64]
Membrane nictitans, cornea	Amphotericin B Flucytosine	Topical, subconjunctival injection, PO (dosing NR) Parenteral: 400 mg/kg/24 h given as two divided doses over a period of 3 weeks	Barrow’s goldeneye, common scoter, red-breasted merganser, long-tailed duck (n = NR)	• Efficacy was observed when amphotericin B was given topically, provided the infection was not within the globe, but safety was not reported. Flucytosine given via parenteral routes was efficacious, but no specific route of administration was mentioned. Safety of Flucytosine was reported.	[65]
^d^ Evaluation of antifungal therapy: • Papers collated only studied on *Candida albicans* for the candidiasis infection. Fluconazole was efficacious in sun conure when pulmonary sites were involved, but only one subject was studied. Ketoconazole was efficacious in Lesser-sulphur crested cockatoo when the site of infection was proventriculus, but only one subject was studied. Amphotericin B was efficacious in a range of ornamental ducks when given topically when the site of infection did not involve the globe, but topical administration might not be feasible due to stress from excessive handling. Flucytosine was efficacious and safe in ornamental ducks when the site of infection is the membrane nictitans and cornea, but the parenteral route of administration was not specified. Although the dosing regimen might be efficacious in these species, there is a need to confirm the route of administration for better recommendations to be made. Additionally, the number of subjects in the study was not mentioned, hence the efficacy and safety data may be less reliable. As such, fluconazole, ketoconazole and flucytosine may be efficacious for the respective sites of infection, but more studies with larger sample size are needed to confirm their efficacy.					
** *Cryptococcus neoformans ^e^* **					
Respiratory tract, globe, periocular tissues, and brain	Voriconazole	1 drop of 1% reconstituted topical ophthalmic drops OS q6 h	African grey parrots (n = 1)	• Efficacy in local (intraocular) fungal infections is unclear because of discontinuation within 5 days due to possible toxicity or systemic spread, hence systemic antifungal therapy is required. Presented with ataxia, lethargy, anorexia and lack of ability to perch, which could be due to voriconazole toxicity or worsening disease due to systemic fungal invasion.	[66]
Respiratory tract, globe, periocular tissues, and brain	Terbinafine Fluconazole	PO 20 mg/kg q12 h for 1 week → 15 mg/kg q12 h PO 15 mg/kg q12 h	African grey parrots (n = 1)	• Antifungal therapy may be efficacious depending on site and severity of infection. Despite treatment for 1 month, radiographs still showed severe and progressive exophthalmos and buphthalmias and minimal changes to the coelomic mass. Cytologic examination of intraocular fluid revealed appearance consistent with *Cryptococcus* spp., even though fungal cultures were negative. Though it was highly effective in decreasing the size of the air sac mass, it was not able to reduce the progression of the disease to the eye. No adverse effects were reported.	[66]
Respiratory tract, globe, periocular tissues, and brain	Amphotericin B Terbinafine Fluconazole	Intraocular infusion: 1.5 mg/kg 3 doses into the globe × 4 months PO 15 mg/kg q24 h PO 15 mg/kg q24 h	African grey parrots (n = 1)	• Antifungal therapy was generally inefficacious, despite mild improvements. Even with intraocular infusion of amphotericin B, persistently elevated beta-globulin levels were observed (though improved from the previous measurement), showing continuous acute inflammation and radiographs showed a decrease in size of the coelomic mass. Dosing intervals for terbinafine and fluconazole were increased to q24 h (from q12 h), after which exophthalmia and buphthalmia in the left globe of the eye continued to increase.	[66]
Respiratory tract, globe, periocular tissues, and brain	Amphotericin B	IV 1.5 mg/kg	African grey parrots (n = 1)	• Antifungal therapy was generally inefficacious. Some mild improvements in activity levels and response were present, but the bird eventually presented with seizures and died shortly after recovery, which was suggested to be due to fungal spread to the brain, causing seizures. Histopathological findings revealed *Cryptococcus* spp. in the cerebrum with multifocal areas of haemorrhage. Given the poor penetration of amphotericin B into the cerebrospinal fluid (CSF) [67], insufficient tissue concentrations may have been reached in the CSF, leading to subtherapeutic treatment.	[66]
Respiratory tract, globe, periocular tissues, and brain	Flucytosine Fluconazole	PO 150 mg/kg q12 h PO 15 mg/kg q24 h	African grey parrots (n = 1)	• Antifungal therapy was generally inefficacious. Despite combination therapy, the size of the periocular swelling and swelling on the globe only decreased minimally, even though both flucytosine and fluconazole are able to penetrate into the vitreous and aqueous humours of the eye [67].	[66]
^e^ Evaluation of antifungal therapy: • Interpretations are not statistically significant due to small sample sizes (n = 1), and there are scarce reports of the outcomes of antifungal therapy in different species. More studies on a larger sample size are required to assess the efficacy of such dosing regimens more accurately in these species. *Cryptococcus neoformans* is an opportunistic pathogen which causes disease, mainly in immunocompromised birds due to their unfavourable high body temperatures for the growth of *Cryptococcus* spp. [66], unlike *Cryptococcus gattii*, which causes disease even in immunocompetent birds [66]. Given the severe disease of the African grey parrot at initial presentation, it is possible that the physiological derangements may have altered the disposition of antifungals and pharmacodynamic interactions of the antifungals. Poor treatment outcomes may have been due to deteriorating immune functions and not necessarily inefficacious antifungal therapy. Inefficacy for amphotericin B may have likely arisen from low concentrations at the site of infection. Flucytosine and fluconazole were used in combination to reduce the development of resistance [66]. Despite their wide distribution in tissues (including the eye), they were inefficacious in coping with the fungal load in the eye or preventing systemic spread to other tissues (brain). Safety profiles from these dosing regimens are unclear as symptoms reported may have been due to the clinical deterioration of fungal disease rather than drug toxicity. Recommendations on the dosing regimens to adopt in *Cryptococcus* infections remain inconclusive. Furthermore, *Cryptococcus neoformans* infections seem to be more invasive and result in disseminated infections, unlike *Cryptococcus gattii* spp., which would require potent systemic antifungals [68].					
** *Cryptococcus neoformans var. gattii ^f^* **					
Beak, infraorbital sinus	Fluconazole	PO 8 mg/kg q24 h × 30 days	Goldie’s lorikeet (n = 1)	• Antifungal therapy was inefficacious due to resistance of isolate to fluconazole. Agglutination tests showed *Cryptococcus neoformans var. gattii* to be of serotype B, and all strains were determined to be resistant to fluconazole. Both birds died within 3 weeks of initiation of oral therapy. Intralesional injections of fluconazole were administered with another biotherapeutic (Cryptococcus 30CH), which seemed to eliminate the gelatinous mass, and thus interpretations of the efficacy of fluconazole is inconclusive.	[69]
Choana	Fluconazole	Intralesional injection 8 mg/kg SID × 30 days, → PO 8 mg/kg q24 h × 30 days	Papua lori (n = 1)		[69]
** *Cryptococcus gattii ^f^* **					
Rhamphotheca of lower beak (and underlying bones), lungs, spleen, and brain	Fluconazole	PO 15 mg/kg q12 h	Citron-crested cockatoo (n = 1)	• Cryptococcus gattii VGIIA subtype. Antifungal therapy was generally inefficacious. Amongst the antifungals for susceptibility, fluconazole showed the highest MIC (50% inhibition) of 8 µg/mL. Bird eventually died, and upon necropsy, disseminated infection to various tissues was concluded. As plasma concentration studies were not performed, it is inconclusive if therapeutic concentrations were achieved in these species. Safety was not reported.	[70]
Humerus	Itraconazole	PO 10 mg/kg q24 h	Pesquet’s parrot (n = 1)	• Antifungal therapy was inefficacious. Cryptococcus gattii sp. showed susceptibility to itraconazole (MIC: 0.125 µg/mL) but to fluconazole at a higher MIC (2.0 µg/mL). Itraconazole plasma concentrations were found to be undetectable. Inefficacy was postulated by the authors to be due to use of compounded itraconazole formulations which lack cyclodextrin, which typically facilitates absorption across the gut. Reduced drug absorption could explain the lack of detection of itraconazole in plasma.	[71]
Humerus	Fluconazole	PO 10 mg/kg BID	Pesquet’s parrot (n = 1)	• Antifungal therapy was generally inefficacious. No clinical progression of disease over the few months of therapy, and tracheal lesions reduced in size, but glottal lesions remained. Though fluconazole plasma concentrations measured were above the MIC, bone biopsy revealed persistent cryptococcal infection. Knowledge on the site of infection is crucial, and the plasma concentration is not truly reflective of tissue concentrations.	[71]
Humerus, Glottis, trachea	Fluconazole Flucytosine Fluconazole	PO 10 mg/kg q12 h PO 50 mg/kg q12 h PO 15 mg/kg q24 h	Pesquet’s parrot (n = 1)	• Fluconazole at lower doses + Flucytosine was minimally effective due to eventual increase in Cryptococcus latex agglutination titres despite an initial decrease. This change in titre values may have been due to resistance. Furthermore, flucytosine was not detected in plasma (0 µg/mL), which led to discontinuation and initiation of higher doses of fluconazole. Efficacy and safety were observed when higher dose of fluconazole was used.	[71]
Glottis, trachea	Amphotericin B	1 mg/kg topical administration via atomisation syringe of reconstituted 5 mg/mL injectable q24 h × 2 weeks → q48 h × 2 weeks	Pesquet’s parrot (n = 1)	• Localised therapy for the glottis and trachea, which reduced the lesions, but there was no complete resolution. Safety was observed.	[71]
^f^ Evaluation of antifungal therapy: • Interpretations are not statistically significant due to small sample sizes (n = 1), and there are scarce reports of the outcomes of antifungal therapy in different species. More studies on a larger sample size are required to assess the efficacy of such dosing regimens more accurately in these species. Serology testing has shown different serotypes of *Cryptococcus* spp. with *Cryptococcus neoformans* and *Cryptococcus gattii* (*Cryptococcus neoformans var. gattii* is the former term for *Cryptococcus gattii*), which can also be further differentiated by molecular typing techniques into different genotypes [70]. The antifungal therapy used for these different subtypes varies and determining the Cryptococcus spp. causing fungal disease may guide decision making in selecting the appropriate antifungal to use. Limited findings with fluconazole have shown resistance to *Cryptococcus gattii* spp. in psittacine birds [69,70], and it is not recommended because it is likely to result in poor therapeutic outcomes. Higher doses of fluconazole (15 mg/kg) showed efficacy in a Pesquet’s parrot but not in the Citron-crested cockatoo. However, this could have been due to a more severe disseminated infection in the Citron-crested cockatoo, especially because the dosing interval was shorter in the Citron-crested cockatoo. Another possible cause for the lack of efficacy in the Citron-crested cockatoo could be due to a more resistant isolate (higher MIC) than that in the Pesquet’s parrot. Fluconazole is reported to reach physiologically privileged sites such as the brain and eye but lower concentrations in the bone [67]. Despite this, efficacy was poor in the Citron-crested cockatoo, where infections involved the brain. Bone biopsy showed *Cryptococcus* spp. present in the bone in the Pesquet’s parrot, but fluconazole was still useful in the treatment of cryptococcal infection. Hence, while site of infection is crucial, the severity of the disease can affect clinical outcomes, and veterinarians should be mindful of the clinical condition of the bird. Future studies should aim to elaborate on the dosing regimens with the severity of the disease in mind.					

BID = twice daily; TID = thrice daily; IM = intramuscular; IV = intravenous; MIC = minimum inhibitory concentration; n = number of (animals); NEB = nebulization; NR = not reported; OS = left eye; PO = oral; QID = four times daily; SC = subcutaneous; SD = single dose; SID = once daily; spp. = species.

**Table 3 jof-09-00810-t003:** Summary of efficacy and safety data in reptile species accompanied by a critical analysis of the study findings.

Site of Infection	Antifungal Used	Dosing Regimen	Species	Evaluation of Dosing Regimens in Relation to Efficacy and Safety	References
** *Chrysosporium anamorph of Nannizziopsis vriesii* ** ** (CANV and related spp.) infections ^a^**					
***Chrysosporium anamorph of Nannizziopsis vriesii* ** **(CANV)**					
Skin	Itraconazole	PO 5 mg/kg q24 h + 1% topical silver sulfadiazine cream q12 h	Boa constrictor (n = 1)	• Lack of efficacy as snake still died after 3 weeks of itraconazole treatment. Safety was not reported.	[72]
Skin	Itraconazole	PO 5 mg/kg q24 h	Coastal bearded dragon (n = 1)	• Lack of efficacy as bearded dragon was euthanised due to persistent systemic mycoses, with several granulomas indicating the presence of fungal infections. Safety was not reported.	[73]
Skin	Itraconazole	PO 5 mg/kg q24 h	Inland bearded dragon (n = 7)	• Possibly efficacious as the fungus was not re-isolated from the dermal lesions after approximately 4 weeks in 2/7 of the bearded dragons. 5/7 of the other bearded dragons did not survive, and upon post-mortem culture, CANV was cultured in 2/5 of these bearded dragons. Possible drug toxicity was apparent.	[74]
Skin	Itraconazole	PO 5 mg/kg q48 h × 14 days + 20 mg/kg SC ceftazidime q72 h × 12 days	Inland bearded dragon (n = 1)	• Efficacy was observed. Safety was not reported.	[35]
Skin	Itraconazole	PO 10 mg/kg q24 h × 6 weeks + 0.125% chlorhexidine topical solution	Inland bearded dragon (n = 1)	• Lack of efficacy was observed. Possible drug toxicity was apparent.	[35]
Skin	Itraconazole	PO 10 mg/kg q24 h × 10 weeks + daily baths in dilute povidone-iodine solution	Inland bearded dragon (n = 1)	• Lack of efficacy was observed. Post-mortem examination revealed an extensive ulcerative dermatitis of the ventral abdomen and a focal hepatic granuloma. Bacterial rods and fungal hyphae were found within the granuloma.	[35]
Systemic	Itraconazole	PO 10 mg/kg q24 h × 21 days	Jewel chameleon (n = 1)	• Lack of efficacy was observed. Safety was observed during the course of treatment, but the duration of therapy was too short for toxicity to be observed.	[75]
Skin	Itraconazole	PO 10 mg/kg q24 h × 21 days	Parson’s chameleon (n = 1)	• Possibly efficacious despite the general loss of condition, but the chameleon’s overall clinical condition improved months later. However, the chameleon still died a year later. Autopsy indicated that death was due to cholecystitis and septicaemia, and no evidence was mycosis was seen. Possible drug toxicity was apparent.	[75]
Skin	Voriconazole	PO 10 mg/kg q24 h	Giant girdled lizard (n = 1)	• Efficacy was observed. Voriconazole had the lowest MIC upon antifungal susceptibility testing (0.25µg/mL) amongst the other antifungals tested. Safety was observed.	[36]
Skin	Voriconazole	PO 10 mg/kg q24 h	Inland bearded dragon (n = 7)	• Efficacy was observed. Possible drug toxicity was apparent.	[74]
*Chrysosporium* spp.					
Skin	Ketoconazole	PO 20 mg/kg q24 h + 2% chlorhexidine solution + topical terbinafine	Green Iguana (n = 1)	• Efficacy was observed. Safety was not reported.	[76]
Skin	Ketoconazole	PO 20 mg/kg q24 h × 14 weeks + 2% chlorhexidine solution + topical terbinafine	Green Iguana (n = 1)	• Efficacy was observed. Safety not reported.	[76]
*Nannizziopsis guarroi*					
Skin	Terbinafine	PO 20 mg/kg SD	Inland bearded dragons (n = 8)	• Efficacy was inconclusive. Multiple-dosing studies on actual diseased bearded dragons are required to demonstrate efficacy. Safety was observed.	[77]
*Paranannizziopsis australasiensis*					
Skin	Itraconazole	PO 5 mg/kg q24 h × 28 days + topical 1% terbinafine ointment × 21 days	Tuatara (n = 1)	• Efficacy was inconclusive. Safety was not reported.	[78]
Skin	Itraconazole	PO 5 mg/kg q24 h × 29 days	Tuatara (n = 1)	• Efficacy was observed. Safety not reported.	[78]
^a^ Evaluation of antifungal therapy: • Interpretations are not statistically significant due to small sample sizes and scarce reports of the outcomes of antifungal therapy in different species. More studies on a larger sample size are required to assess the efficacy of such dosing regimens more accurately in different reptile species. Complications arise in the evaluation of therapeutic efficacy due to the changing nomenclature of CANV spp. Molecular testing has revealed numerous subgroups within CANV that are yet to be accurately identified [73]. In the past, identification of CANV-related species has also been confused with morphologically similar looking species, such as the *Trichophyton* spp. [73]. Hence, even with case reports showing efficacy for CANV-related infections, it is unclear if the fungal pathogens involved were indeed CANV-related spp. Itraconazole has been widely used, with a typical dose ranging from 5 to 10 mg/kg in different reptile species. Lack of efficacy was observed for most of the species, and possible itraconazole-induced hepatoxicity was observed in bearded dragons and a Parson’s chameleon. However, it did seem to show efficacy in another distinctly related species, the tuatara, though the sample size was small. Voriconazole seems to be much more potent in vitro (based on antifungal susceptibility testing) than itraconazole, and it also shows better therapeutic efficacy in bearded dragons and a giant girdled lizard. Though these findings are preliminary, voriconazole seems to have a better efficacy and safety profile than itraconazole. However, species-specific voriconazole toxicity has been observed and caution should be taken through close monitoring upon initiation of therapy. Although ketoconazole seems efficacious, the small sample size (n = 1) in a single species, the Green Iguanas, the superior pharmacokinetics, and potency of voriconazole is still likely to make it the more favourable antifungal for the treatment of CANV-related infections.					
*Ophidiomyces ophiodiicola ^b^*					
Skin	Itraconazole	10 mg/kg SD per cloaca	Cottonmouth (n = 7)	• Efficacy is inconclusive but predicted to be inefficacious as the therapeutic concentrations of itraconazole and hydroxyitraconazole (bioactive metabolite) in both plasma and tissues were not reached. Multiple-dosing studies on actual diseased snakes are required to demonstrate efficacy. No adverse events were observed.	[79]
Skin	Voriconazole	SC 5 mg/kg SD	Cottonmouth (n = 6)	• Efficacy is inconclusive. Multiple-dosing studies on actual diseased snakes are required to demonstrate efficacy. Even though two of the six the cottonmouths survived with no adverse events, the other four died and presented with a range of symptoms—lethargy, depression, loss of righting reflex, and even torticollis in one of the cottonmouths. This could be suggestive of potential voriconazole-induced neurotoxicity.	[79]
Skin	Voriconazole	SC 10 mg/kg SD	Cottonmouth (n = 1)	• Efficacy is inconclusive. Multiple-dosing studies on actual diseased snakes are required to accurately predict efficacy in cottonmouths. Safety was observed despite given a higher dose of voriconazole, but sample size (n = 1) is too small to make conclusions on the safety profile.	[79]
Skin	Voriconazole	SC osmotic pump 22.2 mg/mL (1.02–1.6 mg/kg/h)	Eastern massasauga (n = 2)	• Efficacy was not observed. However, sample size is too small to make any generalised conclusions on efficacy. Safety was not reported.	[79]
Skin	Voriconazole	SC osmotic pump 10 mg/mL (12.1–17.5 mg/kg/h)	Timber rattlesnake (n = 1)	• Efficacy is inconclusive. Therapeutic efficacy cannot be concluded due to small sample size. Safety was not reported.	[79]
Skin	Terbinafine	2 mg/mL (18 mg total dose) × 30 min via nebulisation	Cottonmouth (n = 7)	• Efficacy was inconclusive. In vitro susceptibility concentration (0.015 µg/mL) was maintained for at least 12 h in four of the nebulized cottonmouths. The author recommends using a dose of 2 mg/mL (16 mg total dose) × 30 min via nebulisation once a day for the treatment of snake fungal disease. However, further studies need to be carried out to ensure that this recommendation is clinically relevant in actual diseased cottonmouths. No adverse effects were observed, but all snakes non-significantly gained weight.	[37]
Skin	Terbinafine	24.5 mg (75–190 mg/kg) SC implant cranial to midbody point	Cottonmouth (n = 7)	• Efficacy was inconclusive. In vitro susceptibility concentration (0.015 µg/mL) was maintained for > 5 weeks in implanted snakes (peak concentration of 100 ng/mL at 3 weeks). The author recommends giving a 24.5 mg subcutaneous implant every 5–6 weeks for the treatment of snake fungal disease. However, further studies need to be carried out to ensure that this recommendation is clinically relevant in actual diseased cottonmouths. No adverse effects were observed, but all snakes non-significantly gained weight.	[37]
Mandible and eye	Ketoconazole	PO 50 mg/kg q24 h	Black ratsnake (n = 1)	• Reported as *Chrysosporium ophiodiicola*. Efficacy was not observed. Safety was not reported.	[80]
**^b^** Evaluation of antifungal therapy: • Interpretations are not statistically significant due to small sample sizes and scarce reports of the outcomes of antifungal therapy in different species. More studies on a larger sample size are required to assess the efficacy of such dosing regimens more accurately in different reptile species. Therapeutic recommendations are unclear given the limited studies. Given that some of the studies were single-dose studies, efficacy cannot be concluded as the disposition of the drug may vary in multiple-dose studies and clinical disease state of the snake. Voriconazole and terbinafine appear to be favourable options given that therapeutic plasma concentrations were reached when using different routes of administrations such as SC osmotic pumps or implants and nebulisation. These are especially critical in the treatment of venomous snakes as oral administration is not favourable given that they feed twice a month and the danger posed to the administrator (s10). SC implants are safer and more practical as the snake can be released back into the wild and be captured back weeks later to monitor the outcomes of the treatment. However, multiple-dose studies are required to supplement the results of the single-dose studies and determine if they are indeed appropriate for the treatment of snake fungal disease. Voriconazole also appears to exhibit species-specific neurotoxicity, in this case the cottonmouths but not the Eastern massasaugas or Timber rattlesnake. Hence, caution needs to be made when using voriconazole in snakes as the mechanism of this toxicity is still unclear. Terbinafine might be favoured given its better safety profile, but the only studies available were in cottonmouths, and it is unclear if the same safety profile will be observed in other snake species. Nevertheless, it is recommended to closely monitor the snake upon drug administration, especially if voriconazole is being used.					

BID = twice daily; TID = thrice daily; IM = intramuscular; IV = intravenous; MIC = minimum inhibitory concentration; n = number of (animals); NEB = nebulization; NR = not reported; OS = left eye; PO = oral; QID = four times daily; SC = subcutaneous; SD = single dose; SID = once daily; spp. = species.

## 6. Pharmacokinetic Variabilities in Avians and Reptiles

### 6.1. Azoles

Voriconazole and itraconazole are the focus of this review due to the more extensive studies done and superior pharmacokinetics over older-generation azoles. The various PK parameters for collated antifungals are presented in Table 4 (Avians) and Table 5 (Reptiles).

#### 6.1.1. Voriconazole

##### Different Extent of Saturable Metabolism

Saturable metabolism of voriconazole has been observed across avian species, such as African grey parrots, common ravens, magellanic penguins and racing pigeons, to different extents (Table 4). At high doses, the rate of CYP-mediated metabolism reaches an upper limit, resulting in a disproportionate increases in AUC due to a decrease in CL. In magellanic penguins, AUC increased more than proportionately by 7.4-fold but only 3.1-fold in racing pigeons, despite the same 2-fold increase in dose. Interestingly, in Hispaniolan Amazon parrots, AUC and C_max_ increased rather proportionately despite the doubling of dose when using non-compartmental pharmacokinetic modelling but showed similar non-linearities when a non-linear mixed effects model was used [81]. Such differences may be attributable to interspecies differences in the metabolic pathways of voriconazole metabolism [59], CYP polymorphism profiles [82] or even levels of CYP2C expression [27] across species. Variations in the CYP2 subfamily expression, a major component of voriconazole metabolism, have also been found to be associated with feeding habits and migratory behaviours of avians, with higher levels found in migratory omnivorous species, which could be evolutionarily conserved due to the diverse environmental conditions [27]. Though clearance data were lacking in reptiles (Table 5), the potential of saturable metabolism cannot be ruled out.

##### Species-Specific Autoinduction of CYP Enzymes

Autoinduction in voriconazole has been observed in some avians and reptiles. This adds to the complexity of dosing because of the opposite effects on C_ss_ and AUC due to an increase in CL. Autoinduction is dose- and time-dependent [83] and consequently observed in multiple-dosing studies. Multiple-dosing studies are unfortunately scarce, and only limited data are available showing a decrease in C_ss_ and AUC, possibly alluding to autoinduction (Table 4 and Table 5). This has been observed in racing pigeons [32], Hispaniolan Amazon parrots [81], inland bearded dragons [74], mallard ducks [84], and Western pond turtles [85]. The increase in CL results in lower half-lives upon multiple dosing, which would subsequently affect dosing intervals.

The autoinduction potential of voriconazole in other species cannot be ruled out, hence future studies should focus on elucidating the species-specific PK of voriconazole in multiple dosing regimens, which is more clinically relevant in the treatment of fungal diseases.

Non-linearities in voriconazole PK complicate dosing regimens, especially for multiple-dosing, as concepts of CL and half-life are lost. Hence, caution needs to be made when dose extrapolating across species as interspecies variabilities may exacerbate the unpredictability of these non-linearities in voriconazole PK, and therapeutic drug monitoring should be considered to avoid subtherapeutic doses or toxicity.

##### Varying V across Species

As voriconazole has a large V, saturable plasma and tissue protein binding may be of clinical significance. African grey parrots and Japanese quail showed opposing effects on V when the dose increased (Table 4). While the reasons are unclear, the decreased V in African grey parrots could be attributed to saturable tissue protein binding, whereas the increased V in Japanese quails could be due to saturable plasma protein binding. V consequently affects the half-life and hence the dosing interval. However, such postulations cannot be confirmed without data on the unbound fraction of voriconazole in both plasma and tissues. To our knowledge, plasma protein binding studies are rarely done in exotic animals.

#### 6.1.2. Itraconazole

##### Formulation Differences Affecting Absorption Parameters

Absorption of itraconazole is heavily influenced by its dissolution rate in intestinal fluid due to its poor water solubility. Commercial formulations are preferred over compounded formulations due to the presence of cyclodextrin, which increases dissolution and hence the extent of absorption. This is reflected by the higher C_max_ and AUC in African penguins with commercial formulations (Table 4). A similar phenomenon is seen in voriconazole, where the presence of suspending agents in commercial formulations increase dissolution rate, resulting in an increased rate of absorption, as reflected by a higher C_max_ and lower T_max_ in African grey parrots (Table 4).

Due to its poor solubility, oral absorption is also affected by gastric pH and the presence of food [86]. Differences in gastric pH in avians [20] could contribute to the varying absorption parameters across species, affecting bioavailability and thus the dose to be administered. Despite the lack of data in reptiles, formulation differences and the effect of food might also be observed. In reptiles, intestinal transit time may vary with health status and anatomical differences. Intestinal transit times are longer in herbivorous reptiles due to their longer gut and may be prolonged further in immunocompromised reptiles [87], thus affecting absorption parameters. Further studies should attempt to address if this is clinically significant.

##### Differences in Half-Life across Species

Differences in half-life may be due to varying V or CL parameters. The half-lives of itraconazole appeared to be fairly similar across avian species within the range of 6–10 h (Table 4), with the exception of the Lesser flamingo (75.7 h). The higher half-life in Lesser flamingos might be due to a higher V or lower CL compared to African penguins. Further studies are warranted to compare their body fat composition or extent of plasma or tissue protein binding, which would affect the V of antifungals and explain the differing half-lives. Half-lives in reptiles, however, varied significantly across cottonmouths (14.92 ± 5.33 h), spiny lizards (48.3 h) and Kemp’s Ridley sea turtles (75 h) (Table 5). The generally longer half-lives in reptiles is possibly due to their lower metabolic rates, resulting in a lower CL. However, this cannot be confirmed due to lack of CL and V data for reptiles.

##### Species-Specific Itraconazole/OH-ITRA Ratio

Itraconazole is converted into a bioactive hydroxy-metabolite (OH-ITRA) with similar in vitro potency as the parent drug [7], making it clinically relevant. The itraconazole/OH-ITRA ratio is known to vary across species [88]. In Kemp’s Ridley sea turtles, the plasma itraconazole/OH-ITRA ratio was found to be significantly higher (16:1), compared to avians in general [88], which could be due to the lower metabolic rates. Within avians, the itraconazole concentrations were higher than OH-ITRA concentrations in tissues and plasma for mallard ducks [39] and red-tailed hawks [89], and higher in Blue-fronted Amazon parrots in plasma (tissue concentrations unknown) [90]. Interestingly, an opposite trend was noticed in Humboldt penguins, where OH-ITRA concentrations were higher than itraconazole concentrations in plasma [91], and in racing pigeons, where OH-ITRA concentrations were consistently higher in tissues but not in plasma [92]. Disparities in the itraconazole/OH-ITRA ratio alludes to the possibility of different metabolic pathways [88] or different CYP isoenzyme expressions across species.

The timepoints used for taking blood samples can skew the measurements of itraconazole/OH-ITRA ratio, especially if insufficient time points are taken. This was the case in Humboldt penguins, where the blood samples taken were only up to 12 h, which was lower compared to most of the other species.

Low itraconazole plasma concentrations might be misleading as veterinarians might increase the dose without taking into account the concentrations of the bioactive OH-ITRA metabolite, which may cause unwanted toxicity. The metabolic pathways of itraconazole are still poorly characterised in avian and reptiles, and further insights on the differences in pathways across species can allow for safer dose extrapolations. Tissue concentration studies are recommended, as the tissue concentrations of itraconazole are more reflective of therapeutic efficacy than plasma concentrations.

#### 6.1.3. Other Azoles

Though fluconazole shows linear kinetics in humans [93], when the dose was doubled in African grey parrots, the AUC increased more than proportionately by 3.4-fold (Table 4). Disproportionate changes in AUC could be due to a decrease in renal CL because fluconazole is mainly excreted unchanged. CL decreased by 1.7-fold, hinting at the possible saturation of the active transporters involved in tubular secretion at higher doses. Fluconazole is known to be a substrate of a brush-border transporter (MDR1) involved in tubular drug secretion. Though tubular secretion is reported to be low, it is unclear if differences in expression of such active transporters or interspecies differences in passive reabsorption play a role in the elimination of fluconazole in African grey parrots [94].

### 6.2. Allylamines

#### 6.2.1. Accumulation in Peripheral Tissues

The use of terbinafine is not reported in the EAF for reptiles, unlike avians, but it is increasingly being used because of its wide safety margin and spectrum of activity. Due to its lipophilic and keratinophilic nature, terbinafine typically has a higher V because of accumulation in the skin and adipose tissues [95]. The increase in V with increasing doses in African penguins suggested an accumulation of terbinafine in the peripheral tissues (Table 4). Surprisingly, an opposite trend was observed in red-tailed hawks. This is possible if saturable tissue binding occurred. Changes in V may have implications on half-lives, which affects dosing regimens. However, plasma and tissue protein binding are rarely studied in exotic species. A further complication arises due to terbinafine showing two distinct compartments, which accounts for the long distribution phase, thus resulting in two apparent half-lives in African penguins and red-tailed hawks (Table 4). The terminal half-life is reflective of the actual elimination half-life as it purely involves the elimination phase. Close attention needs to be given when interpreting these half-lives to avoid overdosing, using the first half-life value instead.

#### 6.2.2. Disparities in Half-Lives

Another riveting finding was noted with regards to the significantly lower half-life of terbinafine in red-eared slider turtles (5.4 h) and Western pond turtles (26.2–27 h) (Table 5). Changes in half-lives could be due to a change in CL or V, but these parameters were unavailable. A possible explanation could be that in the red-eared slider turtles’ study, all turtles were female, whereas for the Western pond turtles, they were all males. The increase in protein levels during active states of vitellogenesis in females might cause an increase in plasma proteins. Terbinafine has high plasma protein binding, and large V could have resulted in a lower fraction unbound and thus a lower V and shorter half-life [7]. In bearded dragons, terbinafine plasma concentrations were measured to be significantly higher in females than in males, possibly due to hyperproteinaemia in reproductively active females [77]. Further investigations on the effect of differences in plasma and tissue proteins on V, and hence half-life, across species are warranted, which would affect the efficacy of terbinafine dosing regimens.

**Table 4 jof-09-00810-t004:** Pharmacokinetic parameters of commonly reported antifungals for avian species.

Species	Formulation	Fed/Fasted State	ROA	Dosing Regimen	C_max_/Css (µg/mL)	T_max_ (h)	CL/F (mL/h/kg)	V/F (L/kg)	Half-Life (h)	AUC_0–∞_ (µg.h/mL)	Ref.
**Azoles**											
**Fluconazole**											
African grey parrot (n = 12)	Injectable	NR	PO	10 mg/kg SD	Cmp: 8.19	Cmp: 5.18	Cmp: 53.39	Cmp: 0.90	Cmp: 11.65	187.31	[96]
African grey parrot (n = 20)	Suspension of tablets	NR	PO	10 mg/kg SD	Cmp: 7.45	Cmp: 5.91	Cmp: 64.71	Cmp: 0.86	Cmp: 9.22	154.55	[96]
African grey parrot (n = 5)	Suspension of tablets	NR	PO	10 mg/kg q48 h × 12 days	NR	NR	NR	NR	11.4–16.3	NR	[96]
African grey parrot (n = 12)	Suspension of tablets	NR	PO	20 mg/kg SD	Cmp: 18.59	Cmp: 9.54	Cmp: 38.25	Cmp: 0.56	Cmp: 10.19	522.92	[96]
African grey parrot (n = 10)	Suspension of tablets	NR	PO	20 mg/kg q48 h × 12 days	NR	NR	NR	NR	11.4–16.3	NR	[96]
Cockatiel (n = 28)	Suspension	NR	PO	10 mg/kg SD	Cmp: 4.94 NCmp: 4.941	Cmp: 3.42 NCmp: 3	Cmp: 66.99 NCmp: 66.62	Cmp: 1.84 NCmp: 1.88	Cmp: 19.01	Cmp: 149.28 NCmp: 150.08	[97]
Cockatiel (n = 15)	Crushed tablets in drinking water	Ad libitum	PO	100 mg/L	Day 3 (0800 HRS): 3.69 ± 1.22 Day 3 (1600 HRS): 4.17 ± 1.96 Day 7 (0800 HRS): 9.53 ± 1.48 Day 7 (1600 HRS): 12.95 ± 4.62	NR	NR	NR	NR	NR	[97]
Cockatiel (n = 6)	Crushed tablets in drinking water	Ad libitum	PO	150 mg/L	Day 3 (1100 HRS): 6.42 ± 0.86 Day 3 (1600 HRS): 11.36 ± 4.27 Day 7 (1100 HRS): 4.78 ± 0.91 Day 7 (1600 HRS): 6.61 ± 1.67	NR	NR	NR	NR	NR	[97]
**Itraconazole**											
African penguin (n = 3)	Commercial formulation (Sporanox)	Fed	PO	7 mg/kg SD	NCmp: 0.75 ± 0.27	NCmp: 3.7 ± 1.5	NR	NR	NCmp: 5.8 ± 1.1	NCmp: 6.23 ± 1.40	[98]
African penguin (n = 3)	Compounded formulation	Fed	PO	7 mg/kg SD	NCmp: 0.35 ± 0.14	NCmp: 2.1 ± 1.6	NR	NR	NCmp: 6.2 ± 2.8	NCmp: 2.31 ± 1.00	[98]
African penguin (n = 9)	Commercial formulation (Itrafungol)	Fed	PO	20 mg/kg SD	1.47	5.06	759.86	NR	8.05	AUC_0-last_: 26.32	[86]
Blue-Fronted Amazon parrot (n = 8)	NR	Fed	PO	5 mg/kg q24 h × 14 days	Day 1: 1.74 Day 14: 1.44	Day 1: 3.7 Day 14: 6.0	NR	NR	Day 1: 6.1 Day 14: 3.7	Day 1: 20.96 Day 14: 20.84	[90]
Blue-Fronted Amazon parrot (n = 8)	NR	Fed	PO	10 mg/kg q24 h × 14 days	Day 1: 2.31 Day 14: 3.43	Day 1: 4.9 Day 14: 6.9	NR	NR	Day 1: 6.2 Day 14: 7.2	AUC_0-last_: Day 1: 32.14 Day 14: 71.91	[90]
Humboldt penguin (n = 15)	Commercial capsules (Sporanox)	Fed	PO	7 mg/kg q12 h × 14 days	NCmp, ITRA: 0.26 ± 0.15 NCmp, OH-ITRA: 0.99 ± 0.39	NR	NR	NR	NCmp, ITRA: 9.09 NCmp, OH-ITRA: 11.24	NCmp, AUC (0–12 h), ITRA: 1.96 OH-ITRA: 10.35	[91]
Japanese quail (n = 12)	10% suspension (nebulised without dilution)	NR	NEB	SD × 30 min	NCmp: 27.49 ± 4.58 µg/g ^a^	NCmp: 4 ^a^	NR	NR	NCmp: 35.8 ^a^	NCmp, AUC_0-last_: 1133.01 µg.h/g ^a^	[99]
Japanese quail (n = 12)	1% suspension	NR	NEB	SD × 30 min	NCmp: 4.14 ± 0.19 µg/g ^a^	NCmp: 4 ^a^	NR	NR	NCmp: 24.7 ^a^	NCmp, AUC_0-last_: 106.69 µg.h/g ^a^	[99]
Japanese quail (n = 18)	10% suspension	NR	NEB	SID for 30 min × 5 days	NCmp: 104.8 ± 10.02 µg/g ^a^	NCmp: 0.167 ^a^	NR	NR	NCmp: 164.1 ^a^	NCmp, AUC_0-last_: 4955.63 µg.h/g ^a^	[99]
Lesser Flamingo (n = 17)	Commercial formulation (Itrafungol)	NR	PO	10 mg/kg SD	NLMEM: 1.69	NLMEM: 4.73	NLMEM: 50	NLMEM: 5.67	NLMEM: 75.71	NLMEM: 192.58	[48]
Mallard duck (n = 15)	Acidified with HCL in orange juice	NR	PO	20 mg/kg SD	Cmp, ITRA: 1.07 Cmp, OH-ITRA: 0.34	Cmp, ITRA: 0.8 Cmp, OH-ITRA: 4	NR	Cmp, ITRA: 17.3 Cmp, OH-ITRA: 38.4	Cmp, ITRA: 7.45 Cmp, OH-ITRA: 6.27	Cmp, ITRA: 12.4 Cmp, OH-ITRA: 4.7	[39]
Mallard duck (n = 15)	Formulated with β-CD *	NR	PO	20 mg/kg SD	Cmp, ITRA: 1.35 Cmp, OH-ITRA: 0.27	Cmp, ITRA: 0.5 Cmp, OH-ITRA: 6.3	NR	Cmp, ITRA: 20.7 Cmp, OH-ITRA: 38.4	Cmp, ITRA: 8.5 Cmp, OH-ITRA: 6.74	Cmp, ITRA: 11.9 Cmp, OH-ITRA: 5.1	[39]
Racing pigeon (n = 48)	Capsule (Sporanox/ Trisporal)	Fasted	PO	10.3 mg/kg SD	1.13 ± 0.44	4	NR	NR	NR	NR	[40]
Racing pigeon (n = 10)	Capsule (Sporanox/ Trisporal)	Fasted	PO	10.3 mg/kg q6 h × 3 days	Css: 3.6 ± 0.9	NR	NR	NR	13.3	NR	[40]
**Voriconazole**											
African grey parrot (n = 12)	In water	NR	PO	6 mg/kg SD	NCmp: 0.54	NCmp: 2	NCmp: 2184.9	NCmp: 3.50	NCmp: 1.11	NCmp: 2.75	[100]
African grey parrot (n = 12)	In water	NR	PO	12 mg/kg SD	NCmp: 1.89	NCmp: 4	NCmp: 1151.4	NCmp: 2.63	NCmp: 1.59	NCmp: 10.42	[100]
African grey parrot (n = 12)	In commercial suspending agent	NR	PO	12 mg/kg SD	NCmp: 3.02	NCmp: 2	NCmp: 679.4	NCmp: 1.05	NCmp: 1.07	NCmp: 17.66	[100]
African grey parrot (n = 12)	In commercial suspending agent	NR	PO	18 mg/kg SD	NCmp: 5.67 Cmp: 5.85	NCmp: 2 Cmp: 3.01	NCmp: 521.3 Cmp: 540.21	NCmp: 1.20 Cmp: 1.14	NCmp: 1.59 Cmp: 1.46	NCmp: 34.53 Cmp: 33.32	[100]
African penguin (n = 18)	Suspension	Fasted	PO	5 mg/kg SD	Cmp: 1.89 ± 0.43	Cmp: 0.4 ± 0.43	NR	NR	Cmp: 10.92 ± 7.75	Cmp, AUC (0–24 h): 13.82 ± 5.53	[101]
African penguin (n = 16)	Suspension	Fasted	PO	5 mg/kg q24 h × 8 days	NR	NR	Cmp: 0.18 ± 0.08 L/h	Cmp: 3.34 ± 1.20 L	Cmp: 10.92	Cmp, AUC (84–108 h): 37.7	[101]
Common raven (n = 8)	Solution	Fed	IV	10 mg/kg SD	NR	NR	Cmp: 280.11	Cmp: 1.59	Cmp: 3.92	Cmp: 35.70	[82]
Common raven (n = 4)	Suspension	Fed	PO	6 mg/kg SD	Cmp: 3.17	Cmp: 1.56	Cmp: 415.16	Cmp: 0.97	Cmp: 0.76	Cmp: 14.45	[82]
Common raven (n = 8)	Suspension	Fed	PO	12 mg/kg SD	Cmp: 11.84	Cmp: 2.74	Cmp: 133.72	Cmp: 0.44	Cmp: 1.59	Cmp: 89.74	[82]
Common raven (n = 4)	Suspension	Fed	PO	24 mg/kg SD	Cmp: 13.46	Cmp: 4.30	Cmp: 152.50	Cmp: 0.69	Cmp: 2.83	Cmp: 157.38	[82]
Gyrfalcon (n = 2)	Suspension of powder	NR	IM	12.5 mg/kg SD	NCmp: 5.06 ± 0.81	NCmp: 0.50	NCmp: 220 ± 20	NCmp, Vss: 2.01 ± 0.85	NCmp: 5.59 ± 2.63	NCmp: 55.94 ± 4.75	[102]
Gyrfalcon × peregrine falcon hybrid (n = 1)	Solution injected in quail meat	Fed	PO	12.5 mg/kg q12 h × 1 week	1.7	Day 2	NR	NR	NR	NR	[49]
Hispaniolan Amazon parrot (n = 15)	Suspension	NR	PO	12 mg/kg SD	NCmp: 2.49 PKM: 2.58 NLMEM: 2.81 ± 0.31	NCmp:1.00 PKM: 1.01 NLMEM: 0.96 ± 0.33	NCmp: 1575.56 PKM: 1691.32 NLMEM: 1989.71 ± 0.24	NCmp: 2.05 PKM: 1.70 NLMEM: 1.65	NCmp: 0.90 PKM: 0.70 NLMEM: 0.71 ± 0.27	NCmp: 7.61 PKM: 7.10 NLMEM: 7.62 ± 1.15	[81]
Hispaniolan Amazon parrot (n = 12)	Suspension	NR	PO	24 mg/kg SD	NCmp: 5.08 PKM: 5.17 NLMEM: 4.95 ± 0.44	NCmp: 2.00 PKM: 1.30 NLMEM: 1.46 ± 0.19	NCmp: 1306.01 PKM: 1471.47 NLMEM: 1414.61 ± 0.94	NCmp: 2.35 PKM: 2.48 NLMEM: 2.13 ± 0.01	NCmp: 1.25 PKM: 1.17 NLMEM: 1.72 ± 0.37	NCmp: 18.38 PKM: 16.31 NLMEM: 23.38 ± 3.78	[81]
Hispaniolan Amazon parrot (n = 6)	Suspension	NR	PO	18 mg/kg q8 h × 11 days	NCmp, C(2 h), Day 0: 2.89 NCmp, C(2 h), Day 10: 0.79	NCmp, Day 0: 2 NCmp, Day 10: 2	NR	NR	NCmp, Day 0: 1.15 NCmp, Day 10: 1.29	NCmp, AUC(2–5 h): Day 0: 4.59 AUC (2–5 h), Day 10: 1.17	[81]
Japanese quail (n = 38)	Suspension	NR	PO	20 mg/kg SD	NCmp: 5.8	NCmp: 2	NR	Cmp: 1.77	NCmp: no value Cmp: 1.37	NCmp: no value	[50]
Japanese quail (n = 38)	Suspension	NR	PO	40 mg/kg SD	NCmp: 6.9	NCmp: 2	NR	Cmp: 6.10	NCmp: 9.11 Cmp: 8.45	NCmp: 89.8	[50]
Magellanic penguin (n = 15)	Suspension	Fed	PO	2.5 mg/kg SD	NCmp: 1.08	NCmp: 2.02	NCmp: 17.18	NCmp: 0.18	NCmp: 7.26	NCmp: 14.55	[59]
Magellanic penguin (n = 30)	Suspension	Fed	PO	5 mg/kg SD	NCmp: 2.59	NCmp: 1.03	NCmp: 4.62	NCmp: 0.22	NCmp: 33.71	NCmp: 108.29	[59]
Mallard duck (n = 6)	Solution	Fasted	IV	10 mg/kg SD	NR	NR	NCmp: 530 ± 96	Cmp, Vc: 1.23 ± 0.35 NCmp, Vss: 1.14 ± 0.39	NCmp: 1.34 ± 0.58 Cmp: 1.44 ± 0.44	NCmp: 19.35 ± 3.19	[84]
Mallard duck (n = 6)	Solution	Fasted	PO	10 mg/kg SD	NCmp: 3.94 ± 0.29	NCmp: 0.77 ± 0.68	NR	Cmp: 1.50 ± 0.48	NCmp: 1.11 ± 0.27 Cmp: 1.00 ± 0.37	NCmp: 11.48 ± 2.61	[84]
Mallard duck (n = 6)	Suspension	Fed	PO	20 mg/kg q24 h × 21 days	NCmp, Day 1: 10.91 ± 2.16 NCmp, Day 21: 9.96 ± 1.60	NCmp, Day 1: 2 ± 1.1 NCmp, Day 21: 1	NR	Cmp, Day 1: 0.80 ± 0.25 Cmp, Day 21: 0.90 ± 0.14	NCmp, Day 1: 1.79 ± 0.67 NCmp, Day 21: 0.85 ± 0.08 Cmp, Day 1: 1.41 ± 0.29 Cmp, Day 21: 0.72 ± 0.08	AUC_0-last_: NCmp, Day 1: 55.83 ± 15.44 NCmp, Day 21: 26.14 ± 5.49	[84]
Mallard duck (n = 12)	Suspension	Fasted	PO	20 mg/kg q24 h × 21 days	NCmp, Day 1: 11.06 ± 1.95 NCmp, Day 21: 8.09 ± 2.63	NCmp, Day 1: 1.33 ± 0.52 NCmp, Day 21: 1.08 ± 0.49	NR	Cmp, Day 1: 0.58 ± 0.1 Cmp Day 21: 1.62 ± 0.94	NCmp, Day 1: 1.01 ± 0.22 NCmp, Day 21: 1.16 ± 0.25 Cmp, Day 1: 0.89 ± 0.11 Cmp, Day 21: 1.11 ± 0.28	AUC_0-last_: NCmp, Day 1: 48.68 ± 6.68 NCmp, Day 21: 29.79 ± 15.70	[84]
Peregrine falcon (n = 2)	Suspension of powder	NR	IM	12.5 mg/kg SD	NCmp: 5.79 ± 0.68	NCmp: 1.00	NCmp: 210 ± 20	NCmp, Vss: 1.75 ± 0.44	NCmp: 5.01 ± 2.81	NCmp: 60.07 ± 6.35	[102]
Racing pigeon (n = 15)	NR	NR	IV	2.5 mg/kg SD	NR	NR	NCmp, CL: 120 Cmp, CL: 120	NCmp, V: 1.05 Cmp, V: 1.11	NCmp: 6.15 Cmp: 6.62	NCmp: 18.48 Cmp: 18.82	[32]
Racing pigeon (n = 15)	NR	NR	IV	5 mg/kg SD	NR	NR	NCmp, CL: 83 Cmp, CL: 86	NCmp, V: 1.42 Cmp, V: 1.41	NCmp: 11.82 Cmp: 11.33	NCmp: 60.23 Cmp: 57.93	[32]
Racing pigeon (n = 24)	NR	NR	IV	10 mg/kg SD	NCmp: 6.41 Cmp: 5.58	NR	NCmp, CL: 76 Cmp, CL: 76	NCmp, V: 1.77 Cmp, V: 1.79	NCmp: 16.18 Cmp: 16.25	NCmp: 131.73 Cmp: 130.74	[32]
Racing pigeon (n = 24)	Suspension of crushed tablets in water	NR	PO	10 mg/kg SD	NCmp: 3.65 Cmp: 3.32	NCmp: 2.00 Cmp: 2.15	NCmp: 180 Cmp: 176	NCmp: 2.44 Cmp: 2.60	NCmp: 9.29 Cmp: 10.32	NCmp: 55.08 Cmp: 57.07	[32]
Racing pigeon (n = 20)	Suspension of crushed tablets in water	NR	PO	10 mg/kg q12 h × 4 days	NCmp, Cmax (mean) = 2.35	NR	NR	NR	NCmp, Day 4: 1.6	NR	[32]
Racing pigeon (n = 20)	Suspension of crushed tablets in water	NR	PO	10 mg/kg q24 h × 3 days	NCmp, Cmax (mean), Day 1: 3.68 NCmp, Cmax (mean), Day 3: 2.42	NR	NR	NR	NR	NR	[32]
Racing pigeon (n = 12)	Suspension of crushed tablets in water	NR	PO	20 mg/kg q24 h × 10 days	NCmp, Cmax (mean), Day 1: 8.62 NCmp, Cmax (mean), Day 9: 9.18	NR	NR	NR	NR	NR	[32]
Racing pigeon (n = 6)	NR	NR	NEB	10 mg/mL 0.9% sodium chloride × 15 min	0.41 ± 0.20	1.5	NR	NR	NR	NR	[32]
Red-tailed hawk (n = 12)	Suspension	Fed	PO (by gavage)	10 mg/kg SD	NCmp: 4.7 ± 1.3	NCmp: 2.0 ± 1.2	NCmp: 410 ± 208	NCmp: 1.6 ± 0.84	NCmp: 2.8 ± 0.67	NCmp: 29.0 ± 9.9	[103]
Red-tailed hawk (n = 8)	NR	Fed	PO	10 mg/kg q12 h × 14 days	NCmp: 4.5 ± 2.7	NCmp: 2.4 ± 1.1	NCmp: 515 ± 197	NCmp: 1.5 ± 0.73	NCmp: 2.1 ± 0.8	NCmp: 26 ± 9.6	[103]
Red-tailed hawk (n = 3)	Suspension of powder	Fed	PO	15 mg/kg SD	NCmp: 6.18 ± 1.59	NCmp: 4.86 ± 1.95	NCmp: 485.16 ± 274.93	NCmp: 1.35 ± 0.41 L ^b^	NCmp: 2.29 ± 1.01	NCmp: 45.70 ± 20.96	[104]
Red-tailed hawk (n = 4)	Suspension of powder	Fasted	PO	15 mg/kg SD	NCmp: 7.23 ± 1.34	NCmp: 2.29 ± 0.76	NCmp: 430.57 ± 188.36	NCmp: 1.18 ± 0.36 L ^b^	NCmp: 2.04 ± 0.62	NCmp: 46.01 ± 11.80	[104]
Saker falcon (n = 3)	Suspension of powder	NR	IM	12.5 mg/kg SD	NCmp: 5.57 ± 0.88	NCmp: 0.79± 0.29	NCmp: 200 ± 40	NCmp, Vss: 1.93 ± 0.06	NCmp: 6.53 ± 1.12	NCmp: 62.75 ± 10.53	[102]
Saker falcon (n = 3), Gyrfalcon × peregrine falcon hybrid (n = 2), Gyrfalcon (n = 1)	Fine powder in sterile water	Fed	PO (by gavage)	12.5 mg/kg q12 h × 2 weeks	1.9–2.4	1	NR	NR	NR	NR	[49]
**ALLYLAMINES**											
**Terbinafine**											
African penguin (n = 10) ^c,d^	Tablets compounded into slurry	NR	PO	3 mg/kg SD	NCmp: 0.1 ± 0.02 Cmp: 0.1 ± 0.02	NCmp: 4.0 ± 0.94 Cmp: 2.7 ± 0.96	NCmp 2600 ± 380 mL/h Cmp: 2600 ± 400 mL/h	37 ± 28.5 Cmp: 37.0 ± 22.90 mg/L NCmp: 37.0 ± 28.50 mg/L	NCmp, 1st: 10.0 ± 4.5 Cmp, 1st: 10.0 ± 4.9 NCmp, 2nd: 121 ± 10 Cmp, 2nd: 123 ± 6	AUC_0–last_: NCmp: 1.2 ± 0.17 Cmp: 1.2 ± 0.12	[105]
African penguin (n = 10) ^c,d^	Tablets compounded into slurry	NR	PO	7 mg/kg SD	NCmp: 0.4 ± 0.11 Cmp: 0.2 ± 0.10	NCmp: 4.0 ± 0.87 Cmp: 1.6 ± 0.90	NCmp: 1600 ± 690 mL/h Cmp: 1900 ± 610 mL/h	Cmp: 37.0 ± 23.80 mg/L NCmp: 40.0 ± 28.10 mg/L	NCmp, 1st: 17.0 ± 4.9 Cmp, 1st: 13.0 ± 4.9 NCmp, 2nd: 136 ± 9.7 Cmp, 2nd: 13.0 ± 9.90	AUC_0–last_: NCmp: 4.3 ± 1.86 Cmp: 3.7 ± 1.12	[105]
African penguin (n = 10) ^c,d^	Tablets compounded into slurry	NR	PO	15 mg/kg SD	NCmp: 0.3 ± 0.05 Cmp: 0.2 ± 0.06	NCmp: 4.0 ± 1.23 Cmp: 2.4 ± 1.33	Cmp: 2800 ± 290 mL/h NCmp: 2100 ± 350 mL/h	Cmp: 68.0 ± 21.60 mg/L NCmp: 52.0 ± 18.60 mg/L	NCmp, 1st: 17.0 ± 5.4 Cmp, 1st: 17.0 ± 4.5 NCmp, 2nd: 131 ± 9.9 Cmp, 2nd: 130 ± 11.1	AUC_0–last_: NCmp: 6.0 ± 1.16 Cmp: 5.4 ± 1.13	[105]
African penguin (n = 10)	Tablets compounded into slurry	NR	PO	15 mg/kg q24 h × 4 days	2.1 ± 0.94	0.8 ± 0.84	0.5 ± 0.71	NR	129.0 ± 6.00	NR	[105]
Common shelduck (n = 7)	Crushed commercial tablet suspension (Lamisil)	NR	PO	60 mg/kg SD	NCmp: 3.99–7.17	NCmp: 1.0–2.0	NR	NR	NCmp: 4.18–8.71	NCmp: 22.73–54.75	[106]
Hispaniolan Amazon parrot (n = 6)	Crushed tablet suspension	Fed	PO	60 mg/kg SD	NCmp: 0.11–0.67	NCmp: 2.0–8.0	NR	NR	NCmp: 8.56–13.51	NCmp: 1.90–4.44	[107]
Hispaniolan Amazon parrot (n = 3)	Crushed tablet solution	NR	NEB	1 mg/mL × 15 min	NCmp: 0.048 ± 0.027	NCmp: 0.14 ± 0.09	NR	NR	NCmp: 0.35± 0.18	NCmp: 0.03 ± 0.02	[108]
Hispaniolan Amazon parrot (n = 3)	Raw powder solution	NR	NEB	1 mg/mL × 15 min	NCmp: 0.20 ± 0.18	NCmp: 0.30 ± 0.38	NR	NR	NCmp: 0.31± 0.38	NCmp: 0.11 ± 0.11	[108]
Red-tailed hawk (n = 10) ^d,e^	Crushed tablets in gelatin capsule (in rat belly meat)	Fed	PO	15 mg/kg SD	NCmp: 0.3 ± 0.24 Cmp: 0.3 ± 0.24	NCmp: 5.4 ± 2.98 Cmp: 5.4 ± 2.98	NCmp: 2400 ± 1460 Cmp: 2300 ± 1460	NCmp: 76.8 ± 38.06 Cmp: 72.0 ± 36.6	NCmp, 1st: 14.7 ± 6.67 Cmp, 1st: 15.0 ± 7.13 Cmp, 2nd: 161 ± 78.2	NCmp: 6.2 ± 3.57 Cmp: 6.7 ± 3.67	[109]
Red-tailed hawk (n = 10) ^d,e^	Crushed tablets in gelatin capsule (in rat belly meat)	Fed	PO	30 mg/kg SD	NCmp: 1.2 ± 0.40 Cmp: 1.2 ± 0.40	NCmp: 3.4 ± 0.96 Cmp: 3.4 ± 0.96	NCmp: 1500 ± 720 Cmp: 1400 ± 720	NCmp: 55.2 ± 17.40 Cmp: 50.1 ± 24.4	NCmp, 1st: 17.5 ± 8.7 Cmp, 1st: 18.2 ± 6.3 Cmp, 2nd: 147 ± 65.6	NCmp: 20.1 ± 9.07 Cmp: 21.6 ± 10.10	[109]
Red-tailed hawk (n = 10) ^d,e^	Crushed tablets in gelatin capsule (in rat belly meat)	Fed	PO	60 mg/kg SD	NCmp: 2.0 ± 0.75 Cmp: 2.0 ± 0.75	NCmp: 5.1 ± 3.50 Cmp: 5.1 ± 3.50	NCmp: 1700 ± 750 Cmp: 1400 ± 750	NCmp: 42.2 ± 25.40 Cmp: 45.5 ± 15.4	NCmp, 1st: 13.3 ± 5.03 Cmp, 1st: 13.3 ± 5.13 Cmp, 2nd: 139 ± 42.0	NCmp: 35.3 ± 15.39 Cmp: 43.2 ± 22.76	[109]

AUC_0-∞_ = area under the plasma concentration–time curve from dosing to infinity; AUC_0-last_ = area under the concentration–time curve from the time of dosing to the last measured concentration after dosing; CL = clearance; CL/F = apparent clearance; Cmax = peak plasma drug concentration; Cmp = compartmental model; Css = steady-state drug concentration; HCL = hydrochloric acid; IM = intramuscular; ITRA = itraconazole; IV = intravenous; NCmp = non-compartmental model; NEB = nebulization; NLMEM = nonlinear mixed effects modelling; NR = not reported; OH-ITRA = hydroxyitraconazole; PKM = 1-compartmental pharmacokinetic model; PO = oral; ROA = route of administration; SD: single dose; SID = once daily; Tmax = time to peak plasma drug concentration; V/F = apparent volume of distribution; Vc = Volume of central compartment; Vss = apparent volume of distribution at steady-state; * β-CD = hydroxypropyl-β-cyclodextrin in water; ^a^: Lung tissue pharmacokinetic parameters of itraconazole in Japanese quails. ^b^: Volume was reported in L and not L/kg. ^c^: CL reported in mL/h and not mL/h/kg and V reported in mg/L instead of L/h. ^d^: 1st half-life reported is the first decline and the 2nd half-life reported is the terminal slope, which denotes the true elimination. ^e^: AUC reported in µg.h/mL/kg instead of µg.h/mL.

**Table 5 jof-09-00810-t005:** Pharmacokinetic parameters of commonly reported antifungals for reptile species.

Species	ROA	Dosing Regimen	Cmax/Css (µg/mL)	Tmax (h)	CL/F (mL/h/kg)	V/F (L/kg)	Half-Life (h)	AUC_0–∞_ (µg.h/mL)	Ref.
**AZOLES**									
**Ketoconazole**									
Gopher tortoise (n = 8)	PO	15 mg/kg q24 h × 3 days	Css(avg): 2.2	NR	18 ± 1.2	5.6 ± 0.7	13.2 ± 1.7	49.7 ± 2.7 ^a^	[110]
Gopher tortoise (n = 14)	PO	30 mg/kg SD	3.39 ± 0.12	10.57 ± 0.82	301.2 ± 5.4	NR	11.57 ± 3.46	99.75 ± 7.52 ^a^	[111]
Gopher tortoise (n = 8)	PO	30 mg/kg q24 h × 3 days	Css(avg): 4.4	NR	18 ± 1.8	5.8 ± 0.8	14.4 ± 2.5	105.2 ± 11.9	[110]
**Fluconazole**									
Kemp’s Ridley sea turtle (n = 21)	SC	21 mg/kg SD	NLMEM: 26.16	NLMEM: 0.79	NLMEM: 10.93	NLMEM: 0.79	NLMEM: 50.35	NLMEM: 1921.17 ^a^	[112]
Loggerhead sea turtle (n = 6)	IV	2.5 mg/kg SD	Cmp: 2.7 ± 0.5	NR	Cmp: 8.2 ± 4.3	Cmp, Vss: 1.38 ± 0.26	Cmp: 132.6 ± 48.7	Cmp: 360.4 ± 172.2	[113]
Loggerhead sea turtle (n = 6)	SC	2.5 mg/kg SD	Cmp: 2.1 ± 0.4	Cmp: 4.8	NR	NR	Cmp: 139.5 ± 36.0	Cmp: 368.7 ± 177.5	[113]
Loggerhead sea turtle (n = 4)	SC	21 mg/kg loading dose, followed by 10 mg/kg q5d	Css(4 h): 16.9 ± 1.2 Css(244 h): 19.0 ± 2.8	NA	NR	NR	143	NR	[113]
**Itraconazole**									
Cottonmouth (n = 7)	Per cloaca	10 mg/kg SD	NCmp: ITRA: 0.47 ± 0.15 OH-ITRA: 0.12 ± 0.05	NCmp: ITRA: 10.86 ± 16.57 OH-ITRA: 56 ± 13.86	NR	NR	NCmp: 14.92 ± 5.33	NCmp: 19.01 ± 17.11	[79]
Inland bearded dragon (n = 7)	PO	5 mg/kg q24 h	Cmax: 0.16–7.8 (mean = 3.64)	NR	NR	NR	NR	NR	[74]
Kemp’s Ridley sea turtle (n = 2)	PO	5 mg/kg q72 h	NCmp: 0.19–0.78	NR	NR	NR	NR	NCmp, AUC_0–last_: ITRA: 20.1 ± 12.3 OH-ITRA: 2.18 ± 1.06	[88]
Kemp’s Ridley sea turtle (n = 3)	PO	10 mg/kg q72 h	NCmp: 0.13–0.98	NR	NR	NR	NR	NCmp, AUC_0–last_: ITRA: 20.6 ± 18.8 OH-ITRA: 1.84 ± 2.08	[88]
Kemp’s Ridley sea turtle (n = 3)	PO	15 mg/kg q72 h	NCmp: 0.81–1.07	NR	NR	NR	NCmp: ITRA: 75 OH-ITRA: 55	NCmp, AUC_0–last_: ITRA: 54.0 ± 11.9 OH-ITRA: 3.19 ± 1.22	[88]
Spiny lizard (n = 35)	PO	23.5 mg/kg (mean) q24 h × 3 days	2.48	98.8	NR	NR	48.3	377.21	[114]
**Voriconazole**									
Giant girdled lizard (n = 1)	PO	10 mg/kg q24 h × 10 weeks	2.81–3.34 (mean: 3.04)	NR	NR	NR	NR	NR	[36]
Inland bearded dragon (n = 7)	PO	10 mg/kg q24 h	≥0.91–14.4 (mean = 5.74)	NR	NR	NR	NR	NR	[74]
Northwestern pond turtle (n = 7)	SC	10 mg/kg SD	NCmp: 41.0 ± 1.96	NCmp: 2	NR	NR	NCmp: 15.2	NCmp: 921.1	[85]
Northwestern pond turtle (n = 7)	SC	10 mg/kg q48 h × 14 days	NCmp: 12.4 ± 2.2	NCmp: 50	NR	NR	NR	NR	[85]
Red-eared slider turtle (n = 12)	SC	10 mg/kg q12 h × 7 days	7.58 ± 5.39	1.29 ± 1.28	NR	NR	NR	NR	[115]
**ALLYLAMINES**									
**Terbinafine**									
Cottonmouth (n = 7)	NEB	18 mg (2 mg/mL TBF HCl solution) × 30 min	NCmp: 0.23 ± 0.14	NCmp: 4 ± 3	NR	NR	NR	NCmp, AUC_0–last_: 1.33 ± 0.98	[37]
Cottonmouth (n = 7)	SC implant	24.5 mg (75–190 mg/kg)	NCmp: 0.20 ± 0.20	NCmp: 229 ± 321	NR	NR	NR	NCmp, AUC_0–last_: 75.6 ± 52.9	[37]
Inland bearded dragon (n = 8)	PO	20 mg/kg SD	NCmp: 0.43 ± 0.34	NCmp: 13 ± 4.66	NR	NR	NCmp: 21.24 ± 12.40	NCmp: 11.36 ± 9.81	[77]
Northwestern pond turtle (n = 18)	NEB	18 mg (2 mg/mL) q24 h × 28 days	Cmax (keratin): 56.25 ± 79.10 µg/g	Tmax (keratin): 16.72 ± 8.01 days	NR	NR	NR	AUC (keratin): 1077 ± 1434 day µg/g	[116]
Northwestern pond turtle (n = 7)	PO	30 mg/kg SD	NCmp: 0.79 ± 0.91	NCmp: 1.8 ± 2.8	NR	NR	NCmp: 26.2 ± 12.7	NCmp, AUC_0–last_: 10.91± 11.34	[117]
Northwestern pond turtle (n = 7)	BEC	30 mg/kg SD	NCmp: 1.02 ± 0.79	NCmp: 14.1 ± 12.3	NR	NR	NCmp: 27.0 ± 7.3	NCmp, AUC_0–last_: 33.02 ± 44.20	[117]
Red-eared slider turtle (n = 6)	PO	15 mg/kg SD	NCmp: 0.202	NCmp: 1.00–4.00 (mean = 1.26)	NR	NR	NCmp: 2.67–9.83 (mean = 5.35)	NCmp: 0.319–7.31 ^a^ (mean = 1.21)	[118]

AUC_0–∞_ = area under the plasma concentration-time curve from dosing to infinity; AUC_0-last_ = area under the concentration-time curve from the time of dosing to the last measured concentration after dosing; BEC = bioencapsulation into an earthworm vehicle; CL/F = apparent clearance; Cmax = peak plasma drug concentration; Cmp = compartmental model; Css = steady-state drug concentration; ITRA = itraconazole; IV = intravenous; NCmp = non-compartmental model; NEB = nebulization; NLMEM = nonlinear mixed effects modelling; NA = not applicable; NR = not reported; OH-ITRA = hydroxyitraconazole; PO = oral; ROA = route of administration; SD: single dose; Tmax = time to peak plasma drug concentration; V/F = apparent volume of distribution; Vss = apparent volume of distribution at steady-state; ^a^: unclear if the AUC calculated was to infinity or till the last point of measurement.

## 7. Evaluation of Current Literature and Future Recommendations

Due to the emergence of fluconazole resistance and its limited spectrum of activity in *Candida* and *Cryptococcus* spp., the use of fluconazole in treating fungal diseases has reduced substantially. Itraconazole and voriconazole have an extended spectrum of activities with a critical place in antifungal therapy against filamentous fungi such as *Aspergillus* spp. [119]. In vitro susceptibility testing of newer generation azoles such as posaconazole are also promising given its superior activity against fungal isolates that exhibit resistance to voriconazole and other triazoles [120].

A major limitation is that the PK studies that were conducted had a small sample size and case reports were also isolated and may not have reached statistical significance. Hence, any conclusions made should be done so with caution as the results may be due to random chance. Furthermore, some of the studies were conducted using critically ill exotic animals, whereas some used healthy animals. This could be a potential confounder as it is challenging to conclude if the treatment failure was due to a severe disease state or lack of antifungal activity. Different compartmental models were also used for the computations of pharmacokinetic parameters, and some reported values from multiple models. In general, the majority of the studies collated used non-compartmental modelling in their PK analysis. Although disparities in values amongst different models do not seem significant for most studies, further studies could be conducted to identify the most appropriate modelling method to be used. Predictors of efficacy using PK parameters are also not well-studied in exotic animals, and future studies should focus on this to allow easier prediction of therapeutic outcomes.

Clinical breakpoints for fungal isolates from exotic animals are not well-established, even though they are increasingly adopted in humans as they are more accurate in predicting clinical success because they take into account the PK-PD of the species and the treatment outcomes. The epidemiological cut-off value (ECV or ECOFF), which is the MIC that differentiates between wild-type and resistant fungal pathogens, is more commonly used in exotic animals as the data can be easily obtained for most species [121]. However, MIC values may vary across species, even for the same fungal isolates, which makes clinical breakpoints the right direction to move towards in achieving optimal therapeutic outcomes for the treatment of fungal diseases.

## 8. Conclusions

In addition to their broad spectrum of activity, voriconazole is likely to be the most efficacious antifungal for treatment of fungal diseases in avians and reptiles, especially for aspergillosis. Itraconazole may be a suitable alternative if cheaper generic formulations are available. However, given the wide array of antifungals available, the site of infection and type of fungal disease need to be considered to ensure optimal therapeutic outcomes. There was a lack of data for fungal diseases other than aspergillosis due to the small sample size or limited studies collated. Tissue concentrations are ideal in determining therapeutic efficacy rather than plasma concentrations, but this may not always be logistically feasible.

Toxicity caused by antifungals were species-specific and more pronounced and variable in voriconazole, from polyuria in African grey parrots to neurological adverse events in cottonmouths. Itraconazole showed mild liver abnormalities in bearded dragons and chameleons. Similar adverse effects have been observed in humans. Terbinafine was generally safe in both avian and reptile species.

PK is an integral component to consider when evaluating the efficacy of dosing regimens in exotic animals. Different formulations of itraconazole have varying impacts on absorption parameters. Non-linearities in voriconazole make dose extrapolations unwise due to the unpredictable effects on half-life and other PK parameters. Hence, using species-specific pharmacological data is the most accurate way of designing dosing regimens for antifungal therapy.

Our review also found specific dosing recommendations for the use of terbinafine in Red-eared slider turtles, Western pond turtles, cottonmouths and inland bearded dragons. Though these were single-dose studies, they provide a good starting point for future multiple-dosing studies, given the promising use of terbinafine which accumulates in peripheral tissues and allows for longer dosing intervals.

## Figures and Tables

**Table 1 jof-09-00810-t001:** Predictors of efficacy for different antifungal classes/drugs.

Drug Class/Drug	Concentration/Time Dependent	Prolonged PAFE ^1^	Best Predictor of Efficacy
Allylamines	Not reported	Not reported	Not reported
Azoles ^2^	Time dependent	Yes	AUC (24 h)/MIC
Echinocandins	Concentration dependent	Yes	Cmax/MIC or AUC/MIC
Flucytosine (5-FC)	Time dependent	No	T > MIC ^3^
Polyenes ^4^	Concentration dependent	Yes	Cmax/MIC

^1:^ PAFE describes the effect of the antifungal at exposures above the MIC. PAFE is typically concentration-dependent and allows for longer dosing intervals as fungal pathogen growth is still suppressed beyond the last exposure to the antifungal. ^2^ Though reported as azoles, most of the data are from studies involving triazoles rather than imidazoles. ^3^ T > MIC refers to the time period whereby drug concentrations are above the MIC of the fungal pathogen.^4^ Mainly referring to the conventional deoxycholate amphotericin B (AmB) and the other liposomal formulations of AmB.

## Data Availability

No new data were created or analysed in this study. Data sharing is not applicable to this article.

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
