# Peer review of "A Critical Review on the Dosing and Safety of Antifungals Used in Exotic Avian and Reptile Species"

_jof, 2023, doi:10.3390/jof9080810_

Round 1
Reviewer 1 Report
I appreciate the opportunity to review this manuscript; it is well-written and engaging. The authors of this manuscript did a review of antifungals used in exotic avian and reptile species.
I just have a few comments:
Line 66: MIC means Minimal Inhibitory Concentration.
Table 2: The legend is missing.
Table 2: The information about the evaluation of antifungal therapy can be a subtitle. This information should be removed from Table 2.
Table 2 can be split, 1 table per each genus. If the specie is the same, you can merge the cells of the first column.
Lines 152 - 153: The reference is missing.
Lines 184, 332, 425: in vitro shall be in italics.
Author Response
Line 66: MIC means Minimal Inhibitory Concentration. Amended.
Table 2: The legend is missing. Amended.
Table 2: The information about the evaluation of antifungal therapy can be a subtitle. This information should be removed from Table 2. We thank the reviewer for the suggestion to remove this evaluation from Table 2. In Table 2, there were 8 overarching critical analysis of the findings from the studies presented in the preceding fungal pathogens presented. We feel that placing this analysis within the Table would facilitate the reader to have an overview on the general conclusions and limitations of the findings. Therefore, we have decided to keep the information within the Table but for clarity, we have updated the Table Title to describe this, and also added superscripted alphabets at the relevant sub-headers in the Table to direct readers to the relevant critical analysis.
Table 2 can be split, 1 table per each genus. If the specie is the same, you can merge the cells of the first column. Thank you for this suggestion. Instead of splitting the Table, we have created a sub-header for each genus so that there is 1 less column now.
Lines 152 - 153: The reference is missing. Amended.
Lines 184, 332, 425: in vitro shall be in italics. Amended.
Reviewer 2 Report
The critical review achieves a well-organized summary of relatively recent investigations of antifungal usage, including their dosing and safety, in exotic birds and reptiles. This manuscript offers practicing veterinarians and research scientists not only a summary of an extensive range of previous investigations, but also insight into trends in usage of the broad range of antifungals in exotic avian and reptile species. Additionally, the authors integrated into their review antifungal agents’ mechanisms of action, pharmocokinetic and pharmocodynamic properties, and insight into current gaps in our knowledge about these properties. This review is a significant contribution to antifungal research and development, and it will serve as a valuable resource for veterinarians treating exotic avian and reptile species struggling with a broad range of mycoses.
Line 131: Change 70 to Seventy
Line 167 and 170 and in Table 2 and throughout manuscript: Change spp. to spp.
Pg. 15: under Evaluation of antifungal therapy, Change Candida albicans to Candida albicans
Pg. 18: Italicize Cryptococcus (in bullet with Cryptococcus 30CH)
Pg. 20: Italicize Cryptococcus (in second bullet)
Line 482: Omit the repeated Antifungal terms “antifungal agent” and “fungistatic agent”
Line 482: In Fungal diseases change aspergillus to Aspergillus, candida to Candida, cryptococcus to Cryptoccus, chrysosporium anamorph to Chrysosporium anamorph, nannizziopsis vriessi to Nannizziopsis vriesii, chrysosporium to Chrysosporium, nannizziopsis to Nannizziopsis, paranannizziopsis to Paranannizziopsis, ophidiomyces to Ophidiomyces
Line 548: Review, consider omitting 222
Author Response
Line 131: Change 70 to Seventy. Amended.
Line 167 and 170 and in Table 2 and throughout manuscript: Change spp. to spp. Amended.
Pg. 15: under Evaluation of antifungal therapy, Change Candida albicans to Candida albicans. Amended.
Pg. 18: Italicize Cryptococcus (in bullet with Cryptococcus 30CH). Amended.
Pg. 20: Italicize Cryptococcus (in second bullet). Amended.
Line 482: Omit the repeated Antifungal terms “antifungal agent” and “fungistatic agent”. Amended.
Line 482: In Fungal diseases change aspergillus to Aspergillus, candida to Candida, cryptococcus to Cryptoccus, chrysosporium anamorph to Chrysosporium anamorph, nannizziopsis vriessi to Nannizziopsis vriesii, chrysosporium to Chrysosporium, nannizziopsis to Nannizziopsis, paranannizziopsis to Paranannizziopsis, ophidiomyces to Ophidiomyces. Amended.
Line 548: Review, consider omitting 222. Amended.